# Monitoring bay-scale ecosystem changes in bivalve aquaculture embayments using flow cytometry

Hannah Sharpe[1]*, Thomas Guyondet[2], Jeffrey Barrell[2], Claude Belzile[3], Christopher W. McKindsey[4], Flora Salvo[5], Anaïs Lacoursière-Roussel[1]*

**1** St. Andrews Biological Station, Fisheries and Oceans Canada, St. Andrews, New Brunswick, Canada, **2** Gulf Fisheries Centre, Fisheries and Oceans Canada, Moncton, New Brunswick, Canada, **3** Institut des Sciences de la Mer de Rimouski, Université du Québec à Rimouski, Rimouski, Québec, Canada, **4** Maurice-Lamontagne Institute, Fisheries and Oceans Canada, Mont Joli, Québec, Canada, **5** Merinov, Dartmouth, Nova Scotia, Canada

* hsharpe@unb.ca (HS); Anais.Lacoursiere@dfo-mpo.gc.ca (ALR)

**Data Availability Statement:** Datasets generated for this study are available in the OBIS system (https://ipt.iobis.org/obiscanada/resource?r=monitoring_bay-scale_bivalve_aquaculture_ecosystem_interactions_using_flow_cytometry).

## Abstract

Bay-scale empirical evaluations of how bivalve aquaculture alters plankton composition, and subsequently ecological functioning and higher trophic levels, are lacking. Temporal, inter- and within-bay variation in hydrodynamic, environmental, and aquaculture pressure complicate plankton monitoring design to detect bay-scale changes and inform aquaculture ecosystem interactions. Here, we used flow cytometry to investigate spatio-temporal variations in bacteria and phytoplankton (< 20 μm) composition in four bivalve aquaculture embayments. We observed higher abundances of bacteria and phytoplankton in shallow embayments that experienced greater freshwater and nutrient inputs. Depleted nutrient conditions may have led to the dominance of picophytoplankton cells, which showed strong within-bay variation as a function of riverine vs marine influence and nutrient availability. Although environmental forcings appeared to be a strong driver of spatio-temporal trends, results showed that bivalve aquaculture may reduce near-lease phytoplankton abundance and favor bacterial growth. We discuss confounding environmental factors that must be accounted for when interpreting aquaculture effects such as grazing, benthic-pelagic coupling processes, and microbial biogeochemical cycling. Conclusions provide guidance on sampling considerations using flow cytometry in aquaculture sites based on embayment geomorphology and hydrodynamics.

## Introduction

Canadian shellfish aquaculture (i.e., bivalve mollusc) production reached 40 358 tonnes in 2022 (https://www.dfo-mpo.gc.ca/stats/aqua/aqua22-eng.html), although there is little comprehensive understanding of bay-scale ecosystem effects. Spatio-temporal variation within and between embayments (e.g., culture density, natural plankton abundance, local hydrodynamics) adds complexity to interpretation of monitoring efforts and predictions of bivalve aquaculture

**Funding:** This project was funded by the Fisheries and Oceans Canada Aquaculture Monitoring Program.

**Competing interests:** The authors have declared that no competing interests exist.

ecosystem impacts. In particular, aquaculture interactions with pelagic resources such as grazing, artificial structure, and nutrient cycling are difficult to differentiate from natural variability within coastal dynamics. As Canadian bivalve aquaculture production focuses on filter-feeders that consume plankton, these interactions are critical to understand ecological functioning with implications for aquaculture management.

Decreases in phytoplankton abundance have been observed at bivalve aquaculture embayments worldwide, e.g. between Denmark and Sweden [1], in the Mediterranean [2], and in Daya Bay, South China Sea [3]. However, a reduction in phytoplankton was not observed in Grande-Entrée Lagoon, Canada [4] and Sonier et al. [5] observed higher phytoplankton biomass accumulation (based on chlorophyll measurements) in St. Peters Bay, a Canadian embayment with one of the highest aquaculture lease coverages in the region. Bivalves retain a variety of particles during filter-feeding, including bacteria, phytoplankton, zooplankton, and detritus. They often preferentially select particles based on particle abundance, size, shape, morphology, motility, toxicity, and nutritional content, that varies as a function of seasonality and bay dynamics [6–10]. However, bivalves must first filter a sufficient quantity of particles before preferential particle selection can occur [7]; capturing particles of increasing size with increasing efficiency until a given species' size threshold has been met [11]. Most bivalves have a 100% retention efficiency for particles larger than 4 μm [12–15]. Sonier et al. [5] reported that cultivated blue mussels (*Mytilus edulis*) have a retention efficiency of 20% ± 2% for pico-phytoplankton (0.2 to 2 μm) and 60% ± 3.5% for nanophytoplankton (2 to 20 μm). The general consensus is that picoplankton are retained at an efficiency of 0 to 30%, depending on bivalve species and life stage [16] and that phytoplankton > 4 μm are their most efficiently retained food source [17–20].

Despite having an important role in marine food webs and nutrient cycling, it is difficult to characterize the overall contribution of heterotrophic bacteria to bivalve diets because they can be digested as free-living cells, bacterial aggregates, or attached to seston. Langdon and Newell [21] reported that the Eastern oyster *Crassostrea virginica* retained free-floating (unattached) bacteria with 5.0% efficiency, while the mussel *Geukensia demissa* retained bacteria with 15.8% efficiency. Several authors have suggested that 20 to 30% of bacteria exist within aggregates exceeding 5 μm in size [22, 23], which would be expected to be retained with 100% efficiency. Aggregates are formed when unicellular and multicellular organisms produce exopolymers that coalesce to form gels [24]. Both oysters and mussels can enhance the concentration of transparent exopolymer particles in the environment, thereby promoting aggregate formation [25, 26]. Due to their carbon to nitrogen ratios (3.5 and 6.6 for bacteria and phytoplankton, respectively), it is thought that although bacteria may not contribute much to the metabolic carbon requirements of bivalves, they may contribute substantially to nitrogen requirements [21, 27, 28].

Flow cytometry is a rapid (< 3 min per sample), highly sensitive method to analyze organisms < 20 μm with relatively low operational costs [29, 30] and has been a globally-useful technology for aquatic research and marine monitoring for decades [31]. The benefits of using flow cytometry to quantify changes in bacteria and phytoplankton abundance per class size, and consequently track changes in aquatic environments, have been well-documented for coastal ecosystems [30, 32–34]. This method has been used to characterize ecosystem dynamics in bivalve aquaculture sites worldwide, with results reported as absolute abundance of cells [3, 35], biomass estimates [1, 36], or both [37, 38]. While abundance and relative abundance informs community structure and biodiversity, and shows better correlation to variations in environmental conditions [39, 40], a limit of flow cytometry is the ability to accurately assess the biovolume, and therefore biomass, of the cells analyzed. Additionally, low taxonomic resolution obtained from flow cytometry only allows for bacteria and phytoplankton quantification

based on particle size, pigmentation, or level of nucleic acid content. This is a limitation of the method since bivalves may alter the relative biomass of certain phyla/species by selective grazing [41]. Biomass and species-specific data therefore have important implications for ecosystem productivity and food availability to higher trophic levels, and must be measured using additional methods such as bulk filtrations, microscopy counts, or imaging technology.

Flow cytometry can be used to distinguish two physiologically and ecologically distinct bacterial groups: cells with low nucleic acid content (LNA) and cells with high nucleic acid content (HNA) [42–46]. LNA bacterial cells have streamlined and small genomes, resulting in lower metabolic rates and less capacity to utilize dissolved organic carbon [47, 48]. They thus thrive in nutrient-limited environments whereas HNA cells thrive in environments with high levels of inorganic nutrients and organic matter [49–52]. The relative abundance of HNA to LNA is therefore an informative indicator of environmental conditions and nutrient availability [53].

Flow cytometry is also used to distinguish between pico- and nano-phytoplankton (0.2 to 2 μm and 2 to 20 μm, respectively) in addition to two broad phytoplankton communities: cyanobacteria and eukaryotes. Different cyanobacteria species contain light-harvesting pigments in the phycobiliprotein family, such as phycocyanin (PC) and phycoerythrin (PE), in differing quantities which can be identified using flow cytometry [54]. PC is an indicator for freshwater cyanobacteria [34] while PE is present in high quantities in marine cyanobacterium *Prochlorococcus* and *Synechococcus*, the two most abundant oxygenic phototrophs on Earth [55]. Eukaryotes within both the pico and nano size fractions are principally members of the classes Chlorophyceae (green algae), Pelagophyceae (heterokont algae), Dinophyceae (dinoflagellates), and Bacillariophyceae (diatoms).

Experimental studies have shown that plankton depletion by mussel grazing negatively impacts the fitness of important Canadian fisheries resources, such as lobster larvae [56]. However, aquaculture ecosystem interactions are still not well-documented from field studies. The present work investigates spatio-temporal variations in bacteria and phytoplankton (< 20 μm) measured using flow cytometry in four Atlantic Canadian bivalve aquaculture embayments. Here, we 1) compared inter-bay late-summer surface bacteria and phytoplankton composition and 2) evaluated within-bay spatio-temporal variations in plankton composition at two deep embayments in Nova Scotia and two shallow embayments in New Brunswick. This work is part of the Aquaculture Monitoring Program, launched by Fisheries and Oceans Canada, which aims at assessing the long-term bay-scale impacts of shellfish (bivalve mollusc) aquaculture on marine ecosystems and helps inform policy and decision-making.

## Methods

Water samples were collected from four Eastern oyster (*Crassostrea virginica*) floating cage aquaculture embayments in Atlantic Canada (Fig 1, Table 1). The embayments were selected to span a range of oceanographic conditions and aquaculture lease coverage, with varying pre-existing knowledge of the hydrodynamics and bivalve aquaculture-environment interactions in each of them. Despite regional differences in geomorphology and hydrodynamics, these embayments have been shown to be ecologically similar at the mesozooplankton level [57]. The quantity of bivalves per lease is often unknown as this is generally proprietary information. Within each embayment, samples were collected at a minimum of three stations to gain a general understanding of spatial dynamics for bacteria and phytoplankton abundance and composition (Fig 1). The deep Nova Scotia embayments (Country Harbour, Whitehead) followed a linear path and had a single point of exchange with the open ocean; stations were therefore labeled as Inner, Mid, and Outer. The shallow New Brunswick embayments

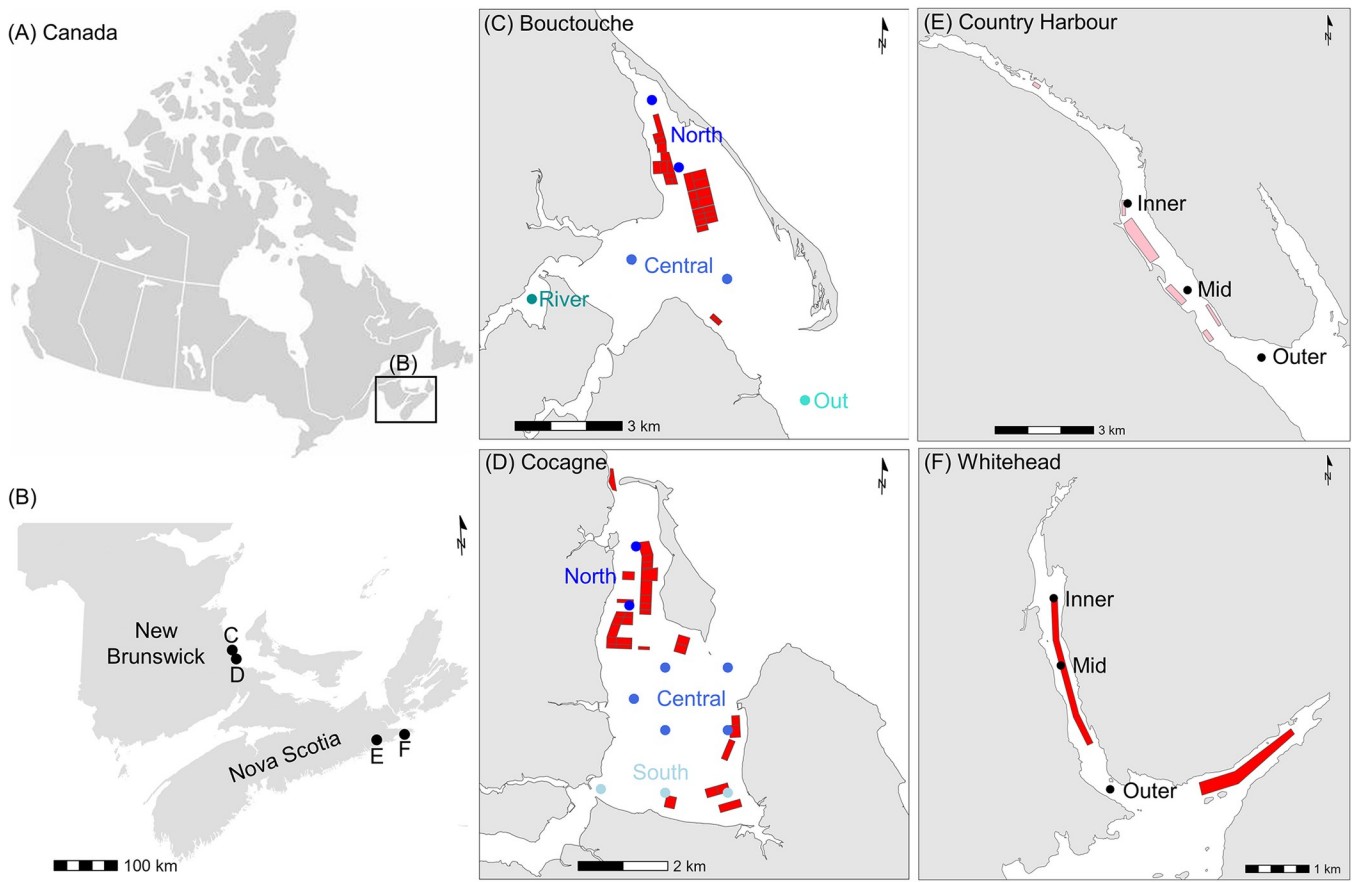

**Fig 1. Location of sampling stations within four bivalve aquaculture embayments in Atlantic Canada.** (A-B) Location of embayments in New Brunswick and Nova Scotia in Atlantic Canada. (C-F) Location of sampling stations in relation to the bivalve aquaculture leases. Red indicates active leases and pink indicates empty leases at time of sampling. Basemap shapefile reprinted from Global Administrative Areas (GADM) database under a CC BY license with permission from GADM (www.gadm.org). Aquaculture lease data provided by the Nova Scotia Department of Fisheries and Aquaculture and the New Brunswick Department of Agriculture, Aquaculture, and Fisheries.

**Table 1. Summary table for bacteria and phytoplankton (0.2 to 20 μm) sampling embayments.**

| Embayment | Bay area (km$^2$) | Lease area (km$^2$) | Cover (%) | Watershed area (km$^2$) | Tidal range (m) | Max depth (m) | Sampling dates | n |
|---|---|---|---|---|---|---|---|---|
| Country Harbour | 10.4 | 0.84 (empty during sampling) | 0.0 during sampling | 235.6 | 0.53–1.87 (Isaacs Harbour, 00535) | 21.9 | Aug 24 (2021) | 18 |
| Whitehead | 1.7 | 0.23 | 13.9 | 63.1 | 0.45–1.80 (Whitehead, 00545) | 14.3 | Aug 25 (2021) | 27 |
| Bouctouche | 28.4 | 1.45 | 5.1 | 890.0 | 0.23–1.28 (St. Thomas de Kent, 01815) | 10.0 | May 25; Jun 7, 22; Jul 8, 21; Aug 3, 16; Sep 1, 13, 27; Oct 25; Nov 9 (2022) | 72 |
| Cocagne | 18.9 | 1.44 | 7.6 | 355.4 | 0.32–1.05 (Cocagne, 01812) | 8.3 | Jun 24, Jul 8, Aug 4, Sep 8 (2021) | 40 |

Cover (%) represents the ratio of the lease area to the bay area. Watershed area is retrieved from New Brunswick Department of Natural Resources and Energy Development (https://hub.arcgis.com/datasets/19d7ac97045442fbac2b0f80d66c53ec_0/about) and the Government of Nova Scotia (https://www.arcgis.com/home/item.html?id=b283921b14214a3889cb47fe0e0aab61). The tidal range (m) is presented as the range between the lower low water mean tide to higher high water mean tide, from the nearest Canadian Hydrographic Service tide station to the sampling stations (https://wla.iwls.azure.cloud.dfo-mpo.gc.ca/stationMgmt). n refers to the number of samples obtained at each embayment.

(Bouctouche, Cocagne) had a more circular coastline therefore stations were labeled according to geographic location and cardinal directions (South, Central, North). At Bouctouche, additional sampling was conducted outside the embayment (Out station) and in the river (River station). All station locations were chosen to sample areas both near and far from leases while keeping an approximately even spacing of stations within embayments.

## Embayment descriptions and sampling design

Country Harbour (10.4 km$^2$) and Whitehead (1.7 km$^2$) are long, narrow channels on the eastern shore of Nova Scotia, Canada (Fig 1, Table 1). These embayments have a semidiurnal tidal influence, and all stations have depths < 16.3 m. Although blue mussels have historically been cultured, Eastern oysters are currently the main cultured bivalve at these embayments, covering 13.9% of the bay area at Whitehead, where 27 samples were collected on 25 August 2021, at three tide phases (high, mid-falling, low) and from three water depths. All leases were empty during sampling at Country Harbour, where 18 samples were collected on 24 August 2021 at two tide phases (high, low) and from three water depths.

Bouctouche (28.4 km$^2$) and Cocagne (18.9 km$^2$) are large, enclosed bays on the eastern coast of New Brunswick, Canada (Fig 1, Table 1). All stations have depths < 4 m, therefore all samples were collected from the surface (1 m below surface). Tidal influence is not considered to be strong in these embayments and was thus not evaluated as a factor. Bouctouche and Cocagne have numerous floating cages containing Eastern oysters within the leases, covering 5.1% and 7.6% of the bay areas, respectively. Sampling at Bouctouche was conducted biweekly from May to November 2022, with water collected from four locations within the embayment, one outside the embayment, and one at the mouth of the river, for a total of 72 samples. Sampling at Cocagne was conducted monthly from June to September 2021 at 10 locations within the embayment, for a total of 40 samples.

## Sample collection

Water samples were collected using a 5 L Niskin bottle and, immediately upon collection, 4.5 mL was combined with 20 μL Grade 1 glutaraldehyde (Sigma; 0.1% final concentration) in cryovials and kept at ambient temperature in the dark for 15 minutes. Subsamples were then placed on ice while in the field and subsequently stored at -80˚C until analysis at Institut des Sciences de la Mer de l'Université du Québec à Rimouski (Canada). At Country Harbour and Whitehead, samples were collected at three depths: surface (S), mid (M), and deep (D). Surface samples were collected 1 m below surface, mid samples were collected approximately halfway down the water column, and deep samples were collected 4 m above seafloor. At Bouctouche and Cocagne, only surface samples were collected due to the shallow water column.

## Flow cytometry

Subsamples for heterotrophic bacteria enumeration were stained with SYBR Green I (Invitrogen) following Belzile et al. [58]. Bacteria (and archaea, collectively referred to here as bacteria) were counted with a CytoFLEX flow cytometer (Beckman Coulter) using the blue laser (488 nm). Green fluorescence of nucleic acid-bound SYBR Green I was measured at 525 nm (525/ 40 nm BP). The cytograms obtained were analyzed using CytExpert v2.3 software and the same regions of the side scatter vs. green fluorescence plots were used to differentiate LNA and HNA bacterial cells.

Subsamples for < 20 μm autotroph abundances (i.e. PE-cyanobacteria, PC-cyanobacteria and autotrophic eukaryotes, collectively referred to as phytoplankton) were analyzed using a CytoFLEX flow cytometer (Beckman Coulter) equipped with blue (488 nm) and red (638 nm)

lasers, following Araújo et al. [59]. Using the blue laser, forward scatter, side scatter, orange fluorescence from PE (582/42 nm BP) and red fluorescence from chlorophyll (690/50 nm BP) were measured. The red laser was used to excite the red fluorescence of PC (660/20 nm BP). Two μm diameter polystyrene microspheres (Fluoresbrite YG, Polysciences) were added to each sample as an internal standard. Pico- (0.2 to 2 μm) and nano-autotrophs (2 to 20 μm) were discriminated based on a forward scatter calibration using algal cultures. Nano-sized autotrophs containing PE or PC were ascribed to nano-cyanobacteria, but may also have been cryptophytes or rhodophytes [60].

## Seston and chlorophyll *a*

At Bouctouche and Cocagne, additional water was collected in conjunction with that used for flow cytometry to allow the analysis of suspended particulate matter (SPM), particulate organic matter (POM), and chlorophyll *a* (chl *a*). For SPM measurements, 1 L duplicates were filtered onto 47 mm Whatman GF/C pre-combusted (500˚C for 4 h in a muffle furnace) and pre-weighed filters (nominal porosity of 1.2 μm) and rinsed with of 0.5 M ammonium formate to reduce salt retention within the filter. The filters were subsequently dried at 70˚C for 24 h, cooled in a desiccator for 2 h, and reweighed for the determination of SPM. Organic matter was then removed by combusting the dried filters at 500˚C for 4 h, then the filters were cooled in a desiccator for 2 h and reweighed for the determination of POM. For chl *a* measurements, duplicate subsamples of 20 and 250 mL were filtered onto 25 mm polycarbonate track-etched (PCTE) filters with nominal porosities of 0.2 and 3.0 μm, respectively. The filters were frozen at -20˚C in the dark until analysis. The pigments retained on the filters were extracted in 10 mL of 90% ice cold acetone and stored at -20˚C in the dark for 24 h. The samples were then warmed to room temperature and the supernatant transferred to a borosilicate tube for fluorescence measurements using the modified fluorometric methods (i.e., narrow bandpass filters to reduce interference from phaeopigments) without acidification [61]. Samples were analyzed using a Turner Trilogy fluorometer calibrated using chl *a* standards from Turner (SKU: 10–850).

## Statistical analysis

Two-dimensional non-metric multidimensional scaling (NMDS) ordinations [62] were used to graphically display spatio-temporal patterns in bacteria (HNA, LNA) and phytoplankton (picoPC-cyanobacteria, nanoPC-cyanobacteria, picoPE-cyanobacteria, nanoPE-cyanobacteria, picoeukaryotes, nanoeukaryotes) community composition (assemblage and abundance). Permutational multivariate analysis of variance (PERMANOVA) was used (number of permutations = 9999) to evaluate variation in plankton communities over time. NMDS ordinations and PERMANOVAs were constructed using a Bray-Curtis dissimilarity matrix of square root transformed abundance data. All analyses were done using the *vegan* package in *R* [63]; the *metaMDS* function was used for NMDS ordinations, and the *adonis2* function was used for PERMANOVAs.

Boxplots were used to visualize spatio-temporal patterns in bacteria and phytoplankton abundance, % HNA (to total bacteria), % picophytoplankton and % cyanobacteria (to total phytoplankton <20 μm). The first, second, and third quartiles, and lines extending from the boxes indicate the minimum and maximum values up to 1.5 times the interquartile range. To determine if each group differed statistically between each other for both medians and distribution, the non-parametric Mann-Whitney U test (*wilcox.test* function in *R*) was used to determine significantly different groups within all factors. This test was chosen due to its ability to compare independent groups that follow a non-normal distribution with uneven and/or

small sample sizes. The Breusch Pagan test (*bptest* function within *lmtest R* package) was subsequently conducted and confirmed that 97.7% of comparisons with equal sample sizes did not meet heteroscedasticity, therefore a correction for type I error was not required for the Mann-Whitney U test [64].

For inter-bay comparisons, surface samples collected from August 4 to September 8 (referred to throughout as late-summer samples) at all four embayments were used. Within-bay statistical analyses conducted for the deep Nova Scotia embayments (Country Harbour, Whitehead) considered the effect of water depth (S, M, D), tide phase (low, high), station (Inner, Mid, Outer), and their interactions as independent variables in bacteria and phytoplankton assemblage composition (dependent variables). When evaluating the effect of tide phase, mid-falling samples were removed to only test low and high tides. Within-bay statistical analysis conducted for the shallow New Brunswick embayments (Bouctouche, Cocagne) considered month (Jun, Jul, Aug, Sep), station (South, Central, North), and their interactions as independent variables in bacteria and phytoplankton assemblage composition. For Bouctouche, only stations within the embayment (Central and North; 2 locations sampled within each station over 7 dates making a total of 14 samples per station) from June 22 to September 13 were used to provide consistency with Cocagne and allow data comparison.

## Results

### Inter-bay comparisons

Late-summer bacteria and phytoplankton composition in surface waters varied significantly between embayments (Fig 2A and 2B, Table 2), with differences primarily between the shallow New Brunswick embayments and the deeper Nova Scotia embayments. The deeper embayments showed no overlap and Country Harbour showed two distinct phytoplankton groups based on riverine (Inner station at high and low tide, as well as Mid station at low tide) and marine (Outer station at high and low tide, as well as Mid station at high tide) influence. The shallow embayments showed similar phytoplankton composition between the Cocagne stations in August and the Bouctouche stations in September.

Late-summer bacteria abundance differed significantly between all four embayments, with the highest and lowest abundances at Bouctouche and Country Harbour, respectively (Fig 3). The relative abundance of HNA bacteria was greater in the shallow New Brunswick embayments than the deeper Nova Scotia embayments. Phytoplankton abundance was significantly higher at Bouctouche relative to the other three embayments, which were statistically similar. Country Harbour had the highest proportion of picophytoplankton and cyanobacteria and, within both the shallow and deeper sets of embayments, relative abundances of both picophytoplankton and cyanobacteria were lower at the embayments with higher lease coverage.

### Spatial and tidal effects at deeper embayments

At Country Harbour, bacteria abundance ranged from $1.4 \times 10^6$ to $2.7 \times 10^6$ cells mL$^{-1}$ and phytoplankton abundance ranged from $0.9 \times 10^5$ to $2.5 \times 10^5$ cells mL$^{-1}$ (Fig 4). HNA represented 42.3 to 55.8% of total bacteria, while picophytoplankton and cyanobacteria ranged from 90.7 to 94.9% and 74.1 to 86.8% of total autotrophic cells, respectively. Station had a significant effect on bacteria and phytoplankton composition (Table 2), with abundances decreasing from Inner to Outer stations while the relative abundance of picophytoplankton increased from the Inner to Outer stations, all of which were not significantly influenced by tide phase or water sample depth (Figs 2C, 2D, 4 and 5). Tide phase did, however, significantly impact the relative abundances of HNA ($p = 0.05$) and cyanobacteria ($p = 0.03$), with greater values during low tide (average 50.9% HNA at low tide, 46.7% HNA at high tide; Fig 5). As well, overall greater

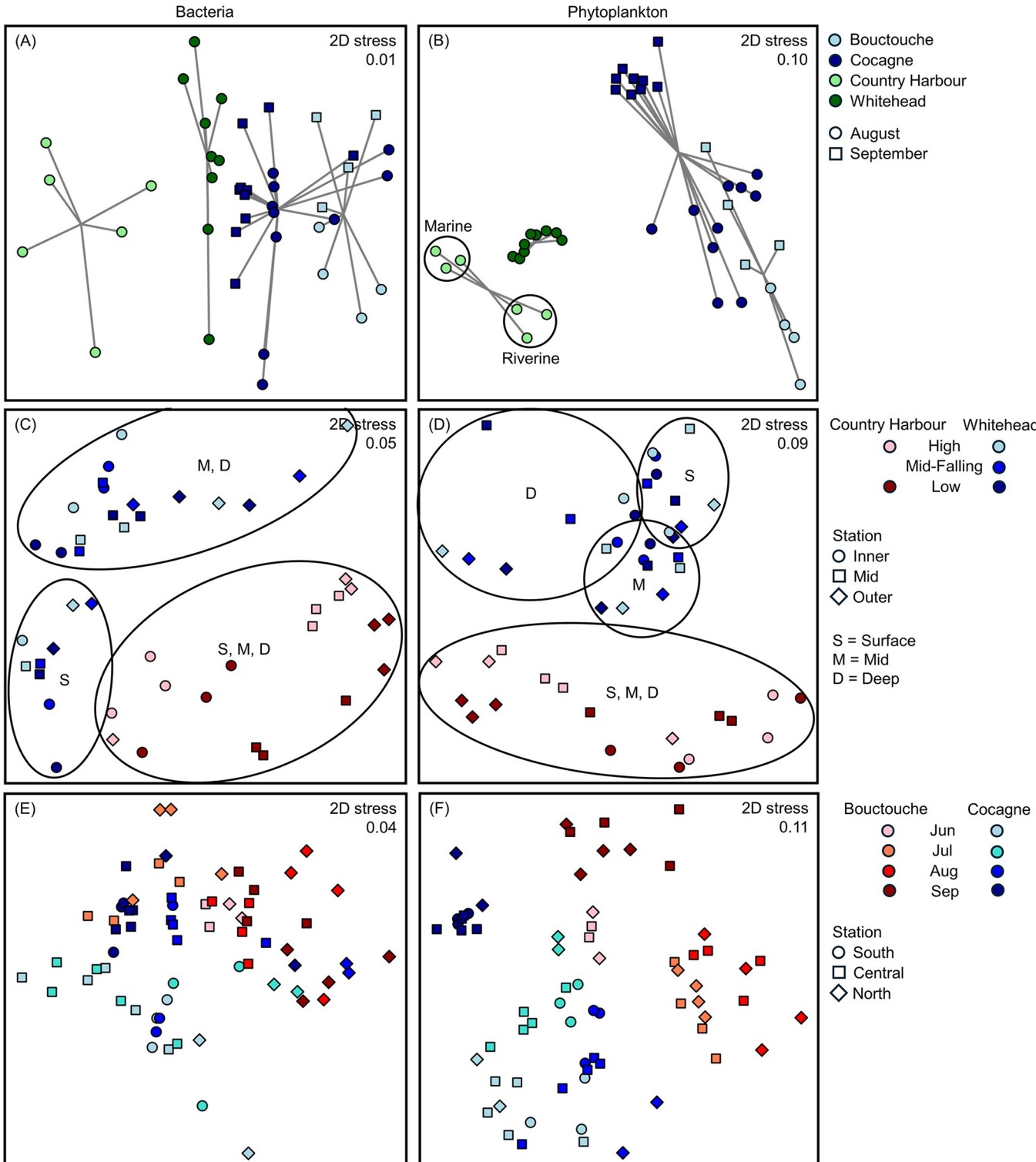

**Fig 2. Two-dimensional non-metric multidimensional scaling ordination (NMDS) showing variation in bacteria and phytoplankton (0.2 to 20 μm) composition assessed using flow cytometry.** (A-B) Inter-bay comparison of late-summer surface samples where color represents embayment while shape represents month. For Country Harbour, groupings based on riverine vs marine influence are shown with circles. (C-D) Inter- and within-bay comparison of Country Harbour and Whitehead where color represents embayment and tide phase and shape represents station. Water sample depth (S = surface, M = mid, D = deep) is indicated by text. (E-F) Inter- and within-bay comparison of Bouctouche and Cocagne from June 22 to September 13 where color represents embayment and sampling month and shape represents station.

**Table 2. Summary of permutational multivariate analysis of variance (PERMANOVA) results based on a Bray-Curtis dissimilarity matrix of square root transformed bacteria and phytoplankton (0.2 to 20 μm) abundances.**

| | | Bacteria | | | | Phytoplankton | | | |
|---|---|---|---|---|---|---|---|---|---|
| | df | SS | $R^2$ | Pseudo-F | P(perm) | SS | $R^2$ | Pseudo-F | P(perm) |
| **Inter-bay comparisons** | | | | | | | | | |
| **Embayment** | **3** | **0.298** | **0.789** | **51.413** | **<0.001** | **0.936** | **0.630** | **22.167** | **<0.001** |
| Residuals | 39 | 0.075 | 0.202 | | | 0.367 | 0.367 | | |
| Total | 42 | 0.373 | 1.000 | | | 1.000 | 1.000 | | |
| **Country Harbour** | | | | | | | | | |
| **Station** | **2** | **0.029** | **0.517** | **10.510** | **0.002** | **0.084** | **0.507** | **11.193** | **0.001** |
| Tide | 1 | 0.004 | 0.076 | 3.098 | 0.097 | 0.010 | 0.062 | 2.739 | 0.113 |
| **Station*Tide** | 2 | 0.006 | 0.111 | 2.260 | 0.145 | **0.027** | **0.159** | **3.521** | **0.052** |
| Residual | 12 | 0.017 | 0.295 | | | 0.045 | 0.272 | | |
| Total | 17 | 0.056 | 1.000 | | | 0.167 | 1.000 | | |
| **Whitehead** | | | | | | | | | |
| **Station** | **2** | **0.023** | **0.314** | **31.973** | **<0.001** | **0.013** | **0.113** | **6.282** | **0.002** |
| **Water sample depth** | **2** | **0.030** | **0.405** | **41.177** | **<0.001** | **0.065** | **0.542** | **30.197** | **<0.001** |
| **Station* Water sample depth** | **4** | **0.014** | **0.193** | **9.805** | **<0.001** | **0.022** | **0.183** | **5.109** | **<0.001** |
| Residual | 18 | 0.006 | 0.088 | | | 0.019 | 0.162 | | |
| Total | 26 | 0.073 | 1.000 | | | 0.119 | 1.000 | | |
| **Bouctouche** | | | | | | | | | |
| **Month** | **3** | **0.046** | **0.670** | **27.451** | **<0.001** | **0.250** | **0.728** | **21.099** | **<0.001** |
| **Station** | **1** | **0.010** | **0.146** | **17.968** | **<0.001** | 0.004 | 0.011 | 0.967 | 0.386 |
| Month*Station | 3 | 0.001 | 0.020 | 0.839 | 0.492 | 0.011 | 0.031 | 0.911 | 0.502 |
| Residual | 20 | 0.011 | 0.163 | | | 0.079 | 0.230 | | |
| Total | 27 | 0.069 | 1.000 | | | 0.344 | 1.000 | | |
| **Cocagne** | | | | | | | | | |
| **Month** | **3** | **0.019** | **0.175** | **8.296** | **<0.001** | **0.421** | **0.734** | **69.178** | **<0.001** |
| **Station** | **2** | **0.056** | **0.512** | **36.475** | **<0.001** | **0.024** | **0.042** | **5.953** | **<0.001** |
| **Month*Station** | **6** | **0.013** | **0.117** | **2.779** | **0.025** | **0.070** | **0.123** | **5.793** | **<0.001** |
| Residual | 28 | 0.022 | 0.196 | | | 0.568 | 0.099 | | |
| Total | 39 | 0.110 | 1.000 | | | 0.572 | 1.000 | | |

df: degrees of freedom; SS: sum of squares; $R^2$: coefficient of variation; Pseudo-F: F statistic by permutation, P(perm): significance by 9999 permutations. Significant effects are shown in bold P(perm) < 0.05).

proportions of HNA were recorded at the Inner (average 50.3%) and Mid (average 49.6%) stations compared to the Outer station (average 46.5%), indicating that greater riverine influence increases HNA cells (Fig 5). None of the parameters measured varied as a function of water sample depth.

At Whitehead, bacteria abundance ranged from $1.6 \times 10^6$ to $3.4 \times 10^6$ cells mL$^{-1}$ and phytoplankton abundance ranged from $0.8 \times 10^5$ to $1.7 \times 10^5$ cells mL$^{-1}$ (Fig 4). HNA represented 32.9 to 55.1% of total bacteria, while the relative abundances of picophytoplankton and cyanobacteria ranged from 70.1 to 87.0% and 55.3 to 80.6%, respectively. Station and water sample depth both had a significant influence on bacteria and phytoplankton composition (Table 2). The Inner and Mid stations were similar for all measured parameters, whereas the Outer station had significantly lower bacteria abundance and higher relative abundances of picophytoplankton, and cyanobacteria. Stations did not differ significantly for phytoplankton abundance or the relative abundance of HNA. Surface samples had significantly higher bacteria abundance, a higher proportion of HNA bacteria, and significantly lower relative abundances of picophytoplankton and cyanobacteria,

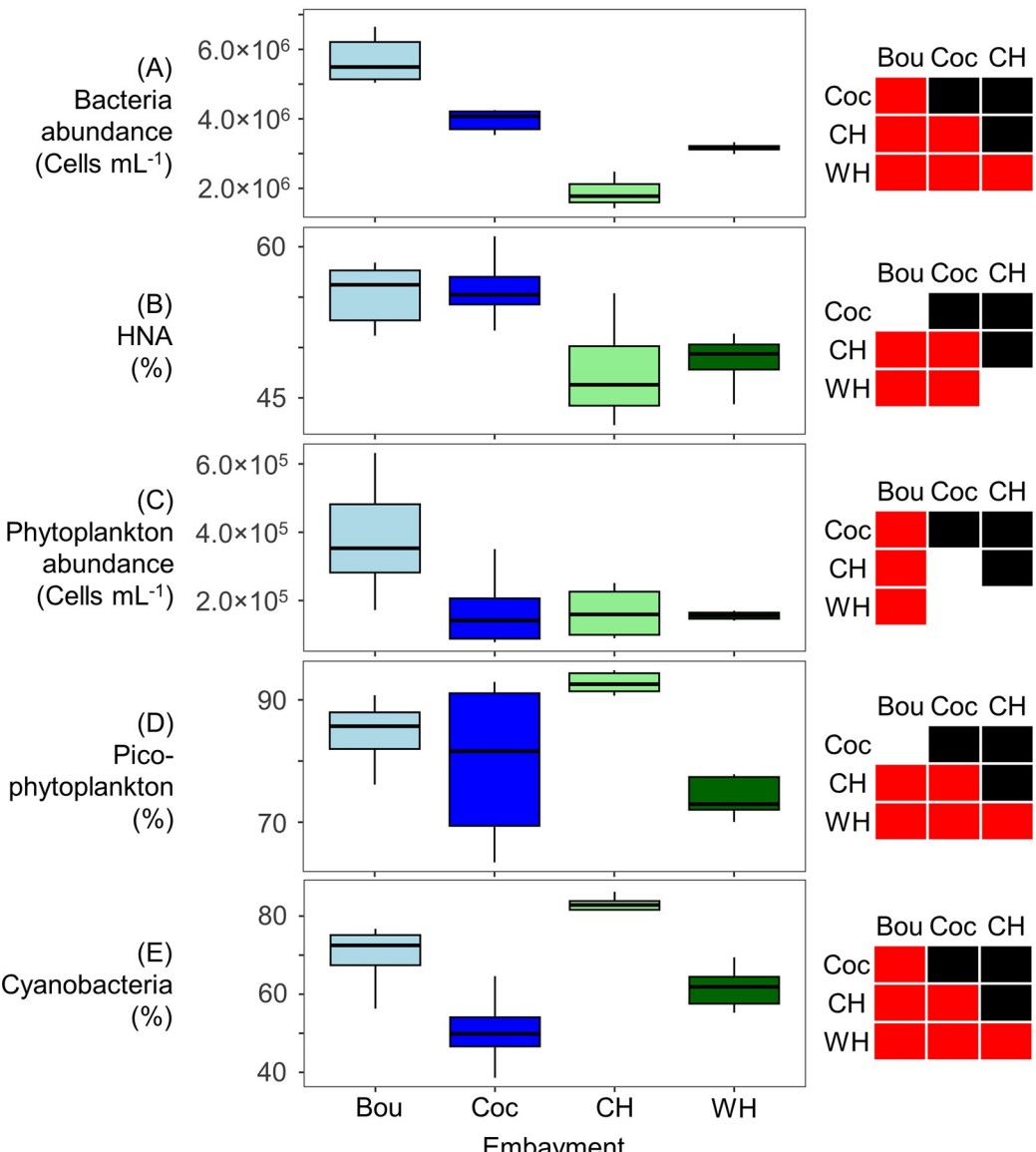

**Fig 3. Boxplots for flow cytometry data (0.2 to 20 μm) of late-summer surface samples from Bouctouche (Bou), Cocagne (Coc), Country Harbour (CH) and Whitehead (WH).** (A) Bacteria abundance, (B) relative abundance of bacteria with high nucleic acid content (HNA) in relation to all bacteria, (C) phytoplankton abundance, (D) relative abundance of picophytoplankton (picoPC-cyanobacteria, picoPE-cyanobacteria, picoeukaryotes) in relation to all phytoplankton cells, (E) relative abundance of cyanobacteria (picoPC-cyanobacteria, nanoPC-cyanobacteria, picoPE-cyanobacteria, nanoPE-cyanobacteria) in relation to all phytoplankton cells. Embayments that differ significantly, based on the Mann-Whitney U tests, are represented by red boxes.

compared to mid and deep samples (Figs 2C, 2D, 4 and 5). Phytoplankton abundance was similar between surface and mid samples but significantly lower in the deeper samples. Tide phase did not significantly influence any of the parameters measured.

## Spatial and seasonal effects at shallow embayments

At Bouctouche, total bacteria abundance ranged from $2.0 \times 10^6$ to $7.7 \times 10^6$ cells mL$^{-1}$ and phytoplankton abundance ranged from $0.4 \times 10^5$ to $6.3 \times 10^5$ cells mL$^{-1}$ (Fig 6). Abundances of

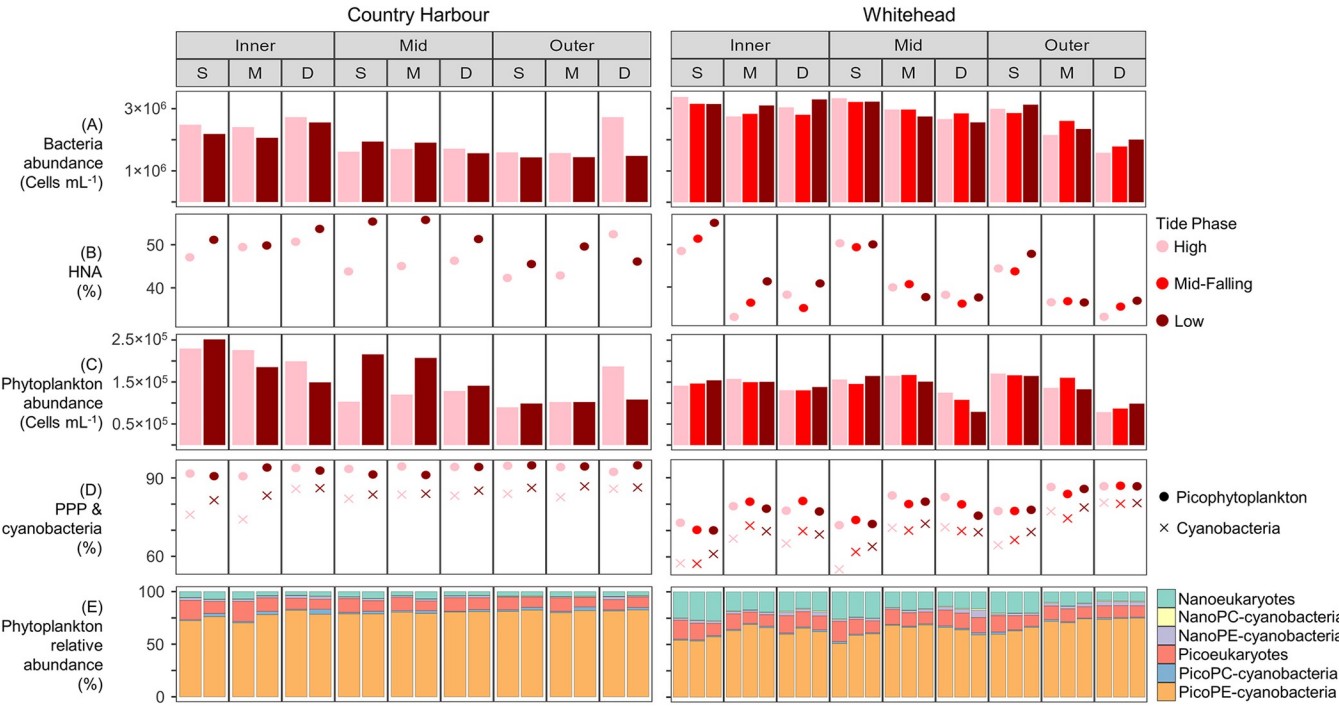

**Fig 4. Flow cytometry data (0.2 to 20 μm) from Country Harbour and Whitehead.** (A) Bacteria abundance, (B) relative abundance of bacteria with high nucleic acid content (HNA) in relation to all bacteria, (C) phytoplankton abundance, (D) relative abundance of picophytoplankton (PPP) and cyanobacteria in relation to all phytoplankton cells, and (E) relative abundance of pico- (0.2 to 2 μm) and nano- (2 to 20 μm) sized eukaryotes, PC-cyanobacteria, and PE-cyanobacteria. For each embayment, samples are grouped by station and water sample depth (S = surface, M = mid, D = deep).

picoPC- and picoPE-cyanobacteria were highest at the River and Out stations, respectively, while abundances of picoeukaryotes were highest within the embayment (Fig 7). Month and station both had a significant effect on bacteria composition, whereas only month significantly affected phytoplankton composition (Table 2). Phytoplankton showed a stronger seasonal trend (Fig 2F) with peak abundance in August, while bacteria abundance was highest in both August and September (Fig 9). The proportion of HNA bacteria ranged from 46.6 to 60.0% and showed high spatio-temporal variability with values > 58% recorded in May (North, River, Central stations), June (River station), August (North, Central, Out stations) and September (North station). Overall, the relative abundance of picophytoplankton ranged from 57.0 to 93.7% and the relative abundance of cyanobacteria ranged from 17.0 to 86.9%. Phytoplankton composition showed a strong seasonal influence, with assemblages dominated by pico- and nano-eukaryotes from May 25 to July 7, by picoPC-cyanobacteria from June 22 to September 13, then by picoPE-cyanobacteria and pico- and nano- eukaryotes from September 27 to November 9. The relative abundance of cyanobacteria was greatest at the River station from May to September, when picoPC-cyanobacteria comprised the majority of phytoplankton cells. Stations within the embayment (Central and North) did not differ significantly for any of the parameters measured (Fig 9).

At Cocagne, bacteria abundance ranged from $2.8 \times 10^6$ to $6.9 \times 10^6$ cells mL$^{-1}$ and phytoplankton abundance ranged from $0.8 \times 10^5$ to $3.5 \times 10^5$ cells mL$^{-1}$ (Fig 6). Abundances of picoPC- and picoPE-cyanobacteria were generally higher near the river and outer embayment, respectively, while abundances of picoeukaryotes were highest at the North station in August (Fig 7). Month, station, and their two-way interaction significantly affected bacteria and phytoplankton composition (Table 2). Both groups showed increasing abundance from June to

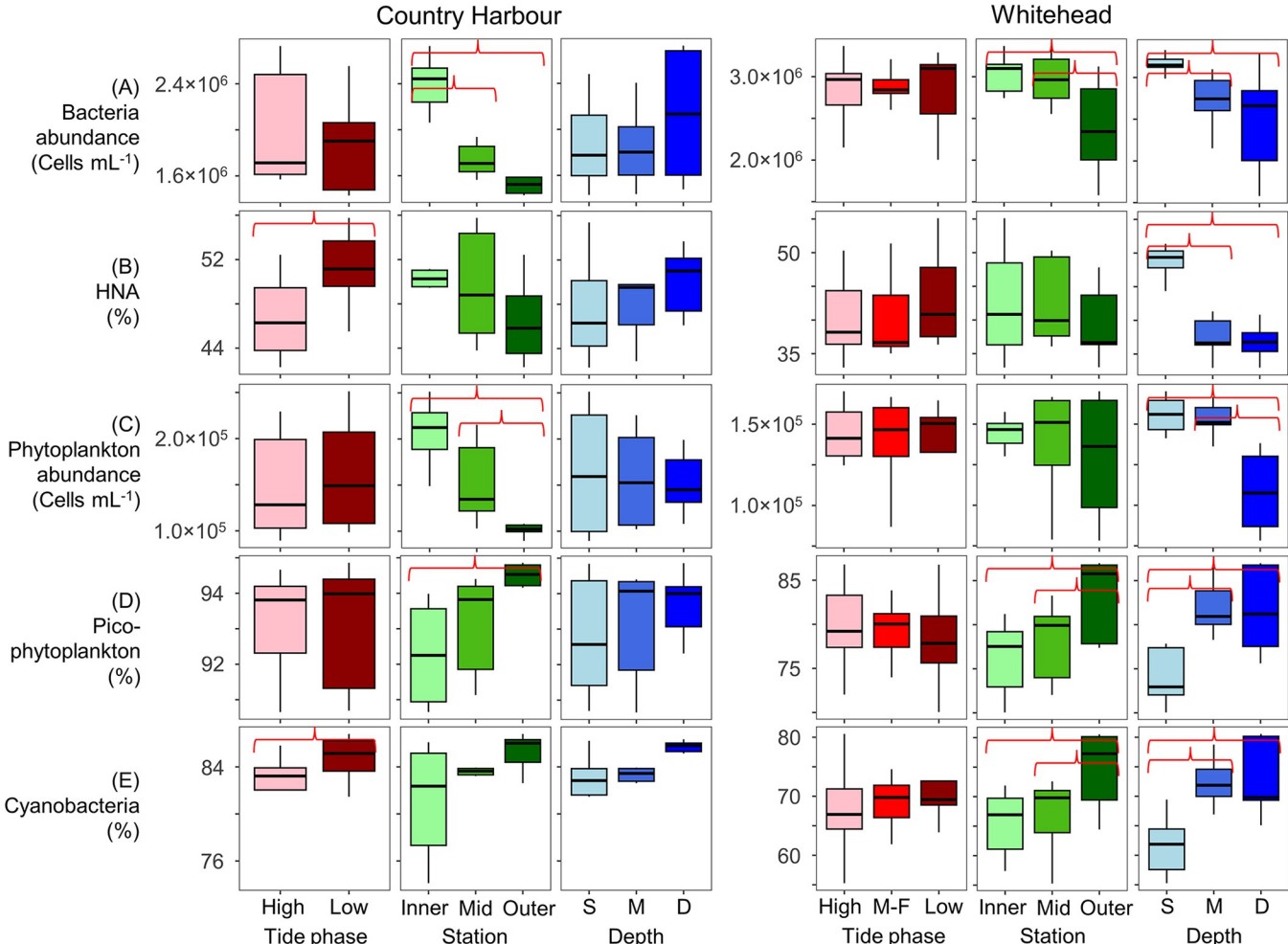

**Fig 5. Boxplots for flow cytometry data (0.2 to 20 μm) from Country Harbour and Whitehead.** (A) Bacteria abundance, (B) relative abundance of bacteria with high nucleic acid content (HNA) in relation to all bacteria, (C) phytoplankton abundance, (D) relative abundance of picophytoplankton (picoPC-cyanobacteria, picoPE-cyanobacteria, picoeukaryotes) in relation to all phytoplankton cells, (E) relative abundance of cyanobacteria (picoPC-cyanobacteria, nanoPC-cyanobacteria, picoPE-cyanobacteria, nanoPE-cyanobacteria) in relation to all phytoplankton cells. For each embayment, boxplots show data distribution between tide phase (M-F = mid-falling), station, and water sample depth (S = surface, M = mid, D = deep). Red brackets highlight significantly different groups based on the Mann-Whitney U test.

August, followed by a decrease in September although the seasonal pattern was less pronounced at Cocagne compared to Bouctouche (Figs 2E, 2F and 6–8). HNA bacteria ranged from 51.7 to 68.4% and consistently decreased from June to September, with the highest proportion of HNA observed at the North station (68.4% in June, 57.9% in September) or near-river at the South station (65.6% in July, 61.0% in August). Phytoplankton composition showed a strong seasonal influence with the relative abundance of cyanobacteria (21.3 to 64.6%) peaking in July and the relative abundance of picophytoplankton (63.4 to 92.9%) peaking in August. Samples were generally similar between stations, except for bacteria abundance, which was significantly greater at the North station (Fig 8).

## Bulk organic matter and chl *a* at shallow embayments

At Bouctouche, SPM ranged from 3.5 to 19.7 mg L$^{-1}$ and POM ranged from 2.0 to 7.3 mg L$^{-1}$ (therefore, 26.0 to 60.2% organics, Fig 9). Total chl *a* ranged from 1.3 to 7.2 μg L$^{-1}$ with the

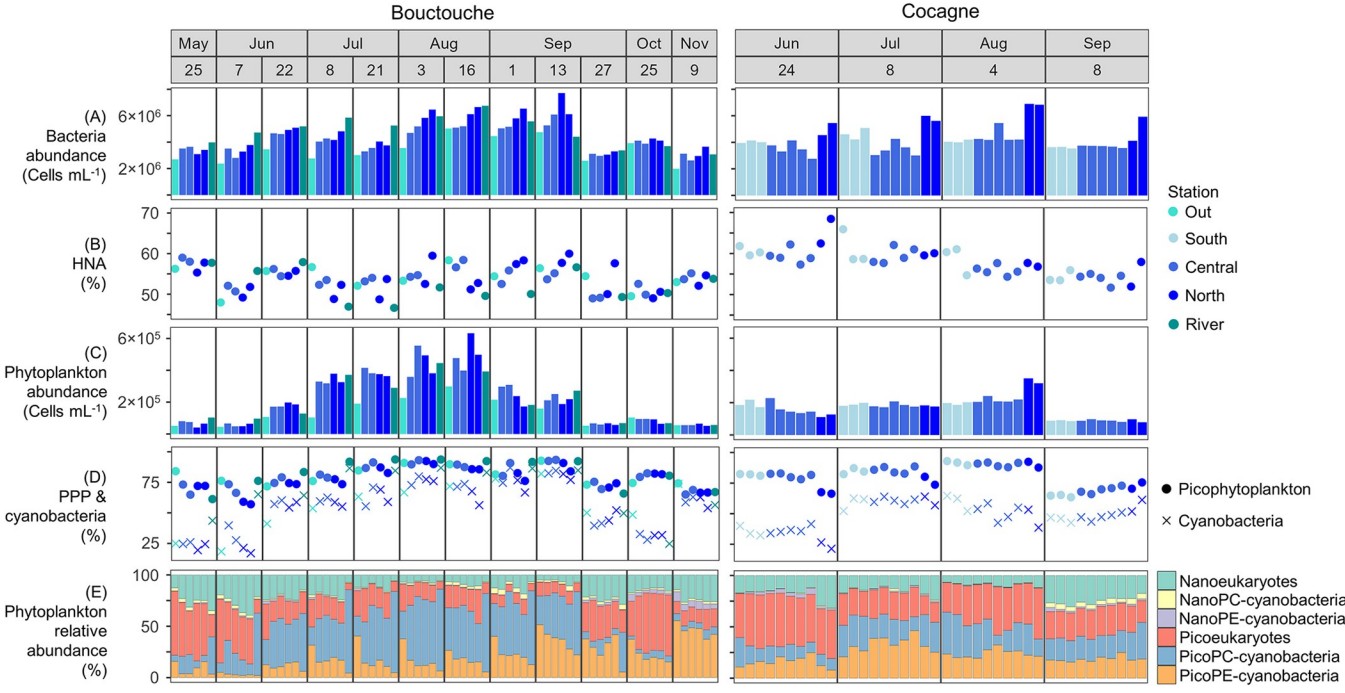

**Fig 6. Flow cytometry data (0.2 to 20 μm) from Bouctouche and Cocagne.** (A) Bacteria abundance, (B) relative abundance of bacteria with high nucleic acid content in relation to all bacteria, (C) phytoplankton abundance, (D) relative abundance of picophytoplankton (PPP) and cyanobacteria in relation to all phytoplankton cells, and (E) relative abundance of pico- (0.2 to 2 μm) and nano- (2 to 20 μm) sized eukaryotes, PC-cyanobacteria, and PE-cyanobacteria.

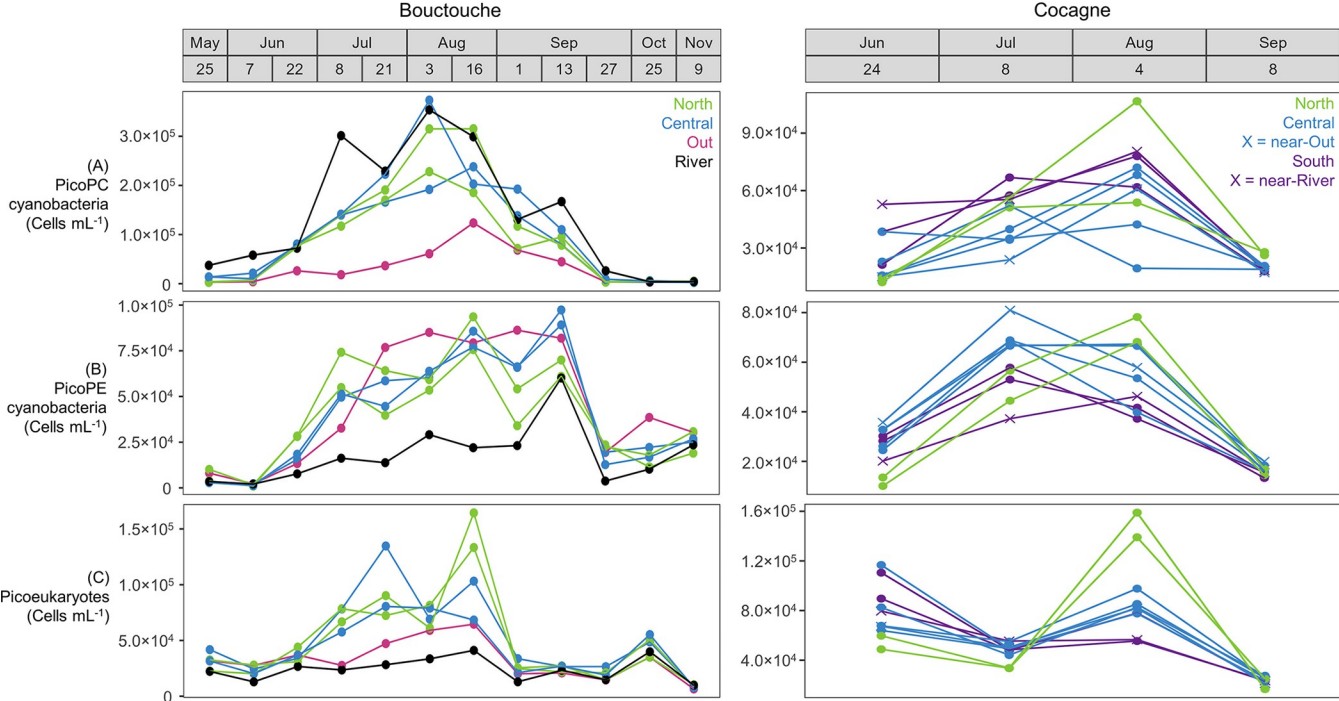

**Fig 7. Flow cytometry picophytoplankton abundance data from Bouctouche and Cocagne.** (A) PicoPC-cyanobacteria, (B) picoPE-cyanobacteria and (C) picoeukaryotes. Colors represent stations.

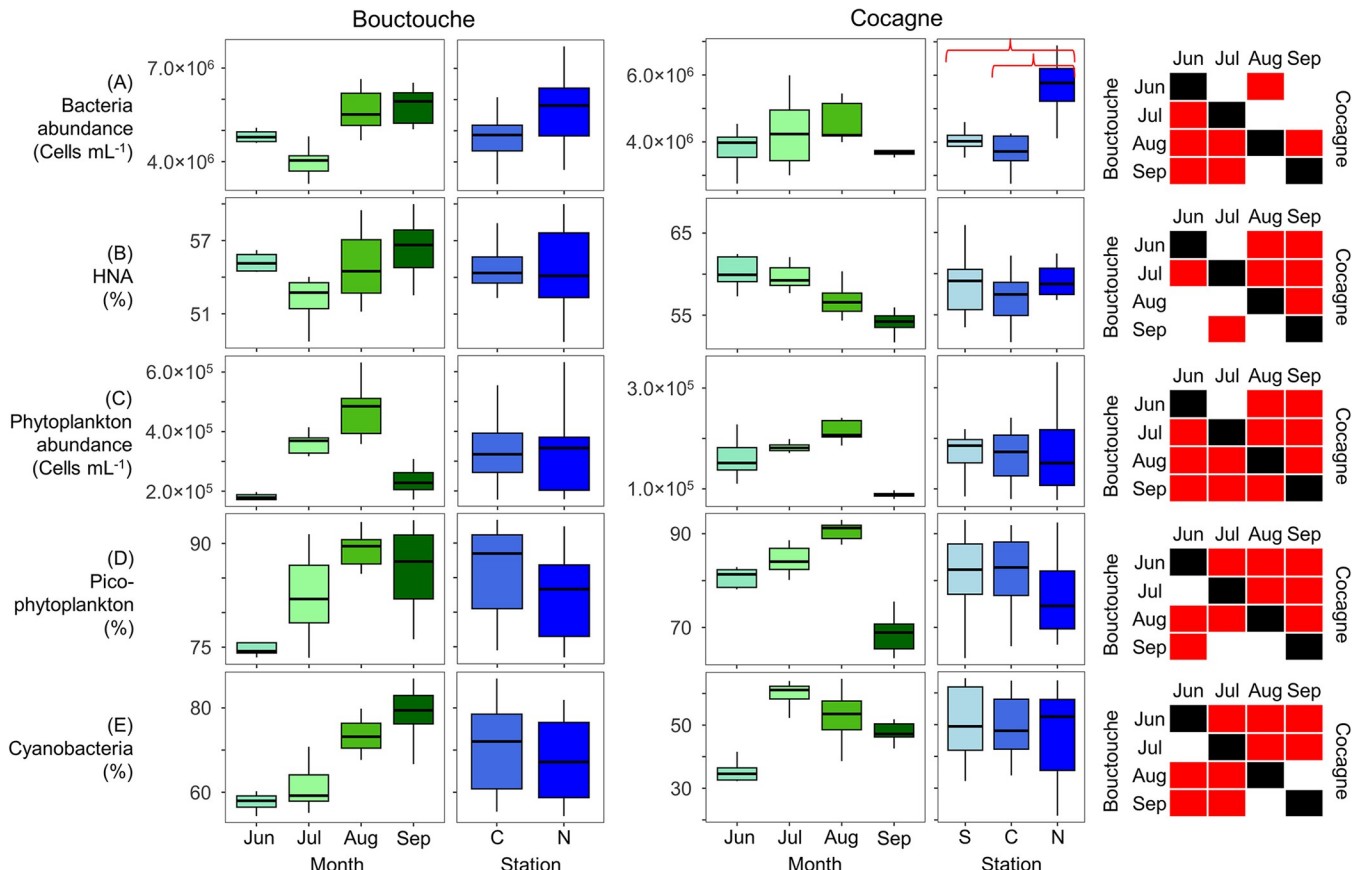

**Fig 8. Boxplots for flow cytometry data (0.2 to 20 μm) from Bouctouche and Cocagne.** (A) Bacteria abundance, (B) relative abundance of bacteria with high nucleic acid content (HNA) in relation to all bacteria, (C) phytoplankton abundance, (D) relative abundance of picophytoplankton (picoPC-cyanobacteria, picoPE-cyanobacteria, picoeukaryotes) in relation to all phytoplankton cells, (E) relative abundance of cyanobacteria (picoPC-cyanobacteria, nanoPC-cyanobacteria, picoPE-cyanobacteria, nanoPE-cyanobacteria) in relation to all phytoplankton cells. For each embayment, boxplots show data distribution between months and stations (S = South, C = Central, N = North). Red brackets highlight significantly different pairwise comparisons between stations based on the Mann-Whitney U tests. Significantly different months for Bouctouche (below diagonal) and Cocagne (above diagonal) are represented using red boxes.

relative concentration of the pico size fraction (0.2 to 3 μm) ranging 27.2 to 88.6%. Stations within the embayment (Central and North) did not differ significantly for any of the parameters measured ([Fig 10]). The months of June to September were generally statistically similar for all parameters measured ([Fig 10]) except for June and July (relative contribution of pico-sized chl *a*), July and September (SPM, POM), and August and September (relative concentration of organics).

At Cocagne, SPM ranged from 3.0 to 17.5 mg L$^{-1}$ and POM ranged from 2.3 to 5.9 mg L$^{-1}$ (therefore, 33.9 to 76.4% organic, [Fig 9]). Total chl *a* ranged from 1.3 to 8.6 μg L$^{-1}$ with the relative concentration of the pico size fraction (< 3 μm) ranging 33.3 to 83.0%. The North station had statistically higher concentrations of SPM, POM, and chl *a*, as well as lower concentrations of pico-sized chl *a* ([Fig 10]). June had the highest concentrations of SPM and POM, while August had the highest percentage of organics.

Linear regressions ([Fig 11]) showed that POM concentrations were correlated to flow cytometry determined plankton ($R^2 = 0.11$, $p = 0.04$) and bacteria ($R^2 = 0.11$, $p = 0.03$) abundance at Cocagne, whereas it was only correlated to phytoplankton abundance ($R^2 = 0.07$, $p = 0.03$) at Bouctouche, although all correlations were generally weak. Correlations between chl *a* and POM concentrations ($R^2 = 0.06$, $p = 0.04$), as well as the relative contribution of

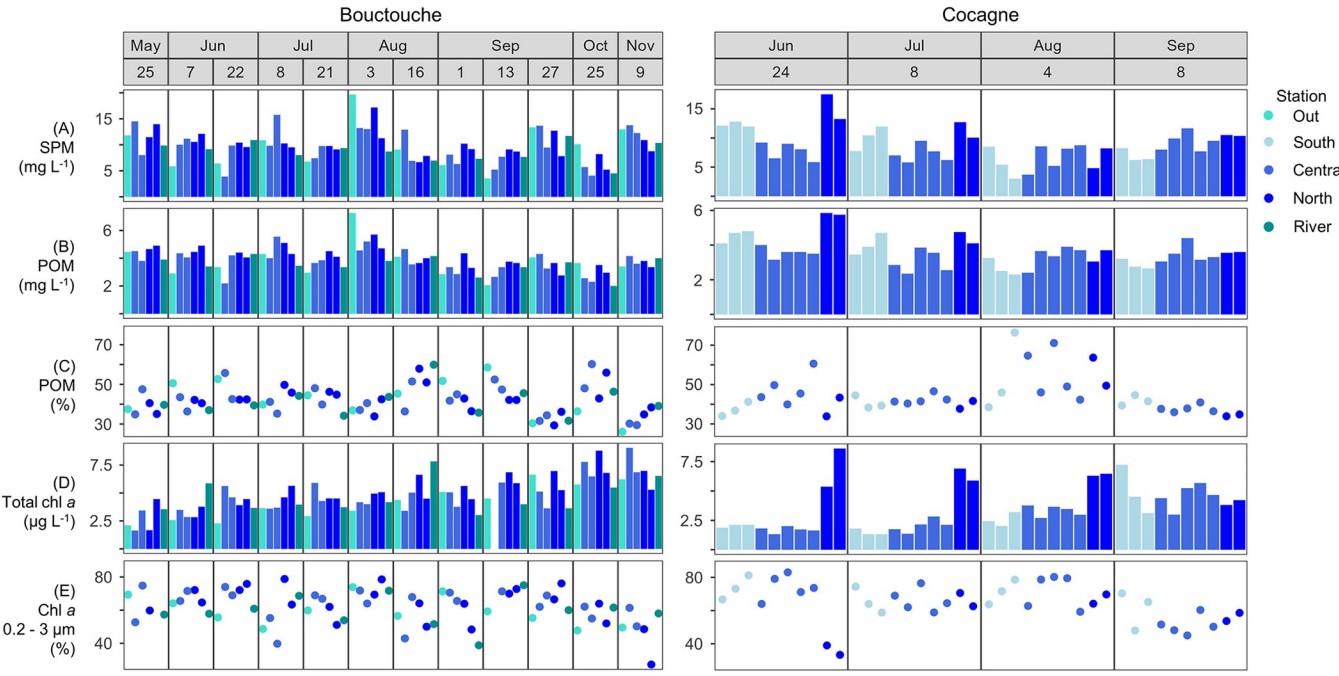

**Fig 9. Bulk organic matter and chl *a* from Bouctouche and Cocagne.** (A) Suspended particulate matter (SPM), (B) particulate organic matter (POM), (C) relative percentage of POM to all SPM, (D) total chl *a* and (E) relative abundance of chl *a* 0.2 to 3 μm (of total chl *a*).

picophytoplankton measured from chl *a* filtrations and cell abundance ($R^2 = 0.34$, $p < 0.001$) were observed at Cocagne but not Bouctouche.

## Discussion

Bacteria and phytoplankton form the base of marine food webs and are the primary food source for larger nano- and micro-grazers (e.g., heterotrophic flagellates, ciliates, copepods) that are in turn grazed by larger zooplankton. Changes in bacteria and phytoplankton can therefore lead to repercussions in the marine food chain, with impacts on metazooplankton and fish [65–67]. For example, experimental studies have shown that plankton depletion by mussel grazing may negatively impact the fitness of important Canadian fisheries resources, such as lobster [56]. However, these dynamics have not been observed and quantified in field studies due to the complexity and dynamics of these ecosystems.

Here, inter- and within-bay patterns in bacteria and phytoplankton composition were observed in conjunction with bay dynamics and bivalve aquaculture. Higher abundances of bacteria were observed in shallow enclosed embayments that experienced greater freshwater and nutrient input. Seasonal patterns in phytoplankton abundance and composition were recorded at both New Brunswick embayments, with stronger seasonal variation observed in 2022 during record high sea surface temperatures and precipitation levels. Depleted nutrient conditions during summer months may have led to the dominance of picophytoplankton abundance and biomass. Picophytoplankton functional groups showed clear spatio-temporal trends, with greatest PC-cyanobacteria (absolute and relative) abundance associated with freshwater influence (River station) whereas PE-cyanobacteria abundance was associated with marine influence (Out station). Absolute picoeukaryote abundance was greatest within the embayment (North, Central and South stations) and varied as a function of seasonality, with particularly clear station-driven differences in August. Although environmental forcings

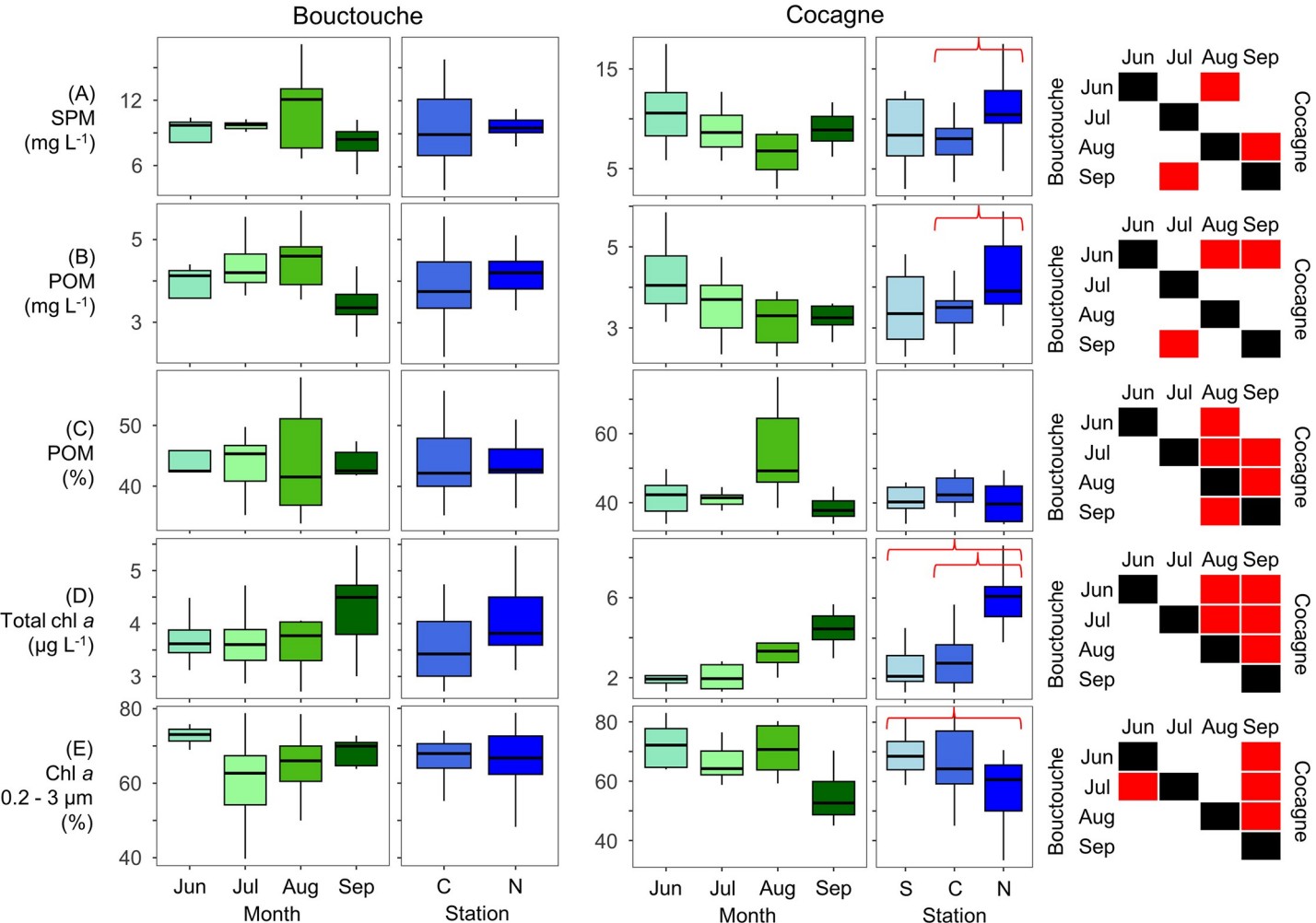

**Fig 10. Boxplots of bulk organic matter and chl *a* from Bouctouche and Cocagne.** (A) Suspended particulate matter (SPM), (B) particulate organic matter (POM), (C) relative percentage of POM to all SPM, (D) total chl *a* and (E) relative abundance of chl *a* 0.2 to 3 μm (of total chl *a*). For each embayment, boxplots show data distribution between months and stations (S = South, C = Central, N = North). Red brackets highlight significantly different pairwise comparisons between stations based on the Mann-Whitney U tests. Significantly different months for Bouctouche (below diagonal) and Cocagne (above diagonal) are represented using red boxes.

appeared to be a strong driver of spatio-temporal trends, results showed that bivalve aquaculture may reduce near-lease phytoplankton abundance and favor bacterial growth.

## Inter-bay dynamics

Bacteria and phytoplankton composition in surface waters differed significantly between embayments, showing similarities between the two shallow, enclosed New Brunswick embayments (Bouctouche, Cocagne), and the two deeper, channel-shaped Nova Scotia embayments (Country Harbour, Whitehead). The shallow enclosed embayments generally had higher bacteria abundance and relative abundance of HNA bacterial cells. HNA cells thrive in waters with high levels of inorganic nutrients and organic matter [49–52], suggesting that the shallow embayments had higher nutrient availability. Phytoplankton abundance was significantly greater at Bouctouche than the other three embayments, which may be driven by nutrient inputs from the watershed area, which is largest at Bouctouche (890 km$^2$) compared to the others (355.4, 235.6 and 63.1 km$^2$ for Cocagne, Country Harbour and Whitehead, respectively).

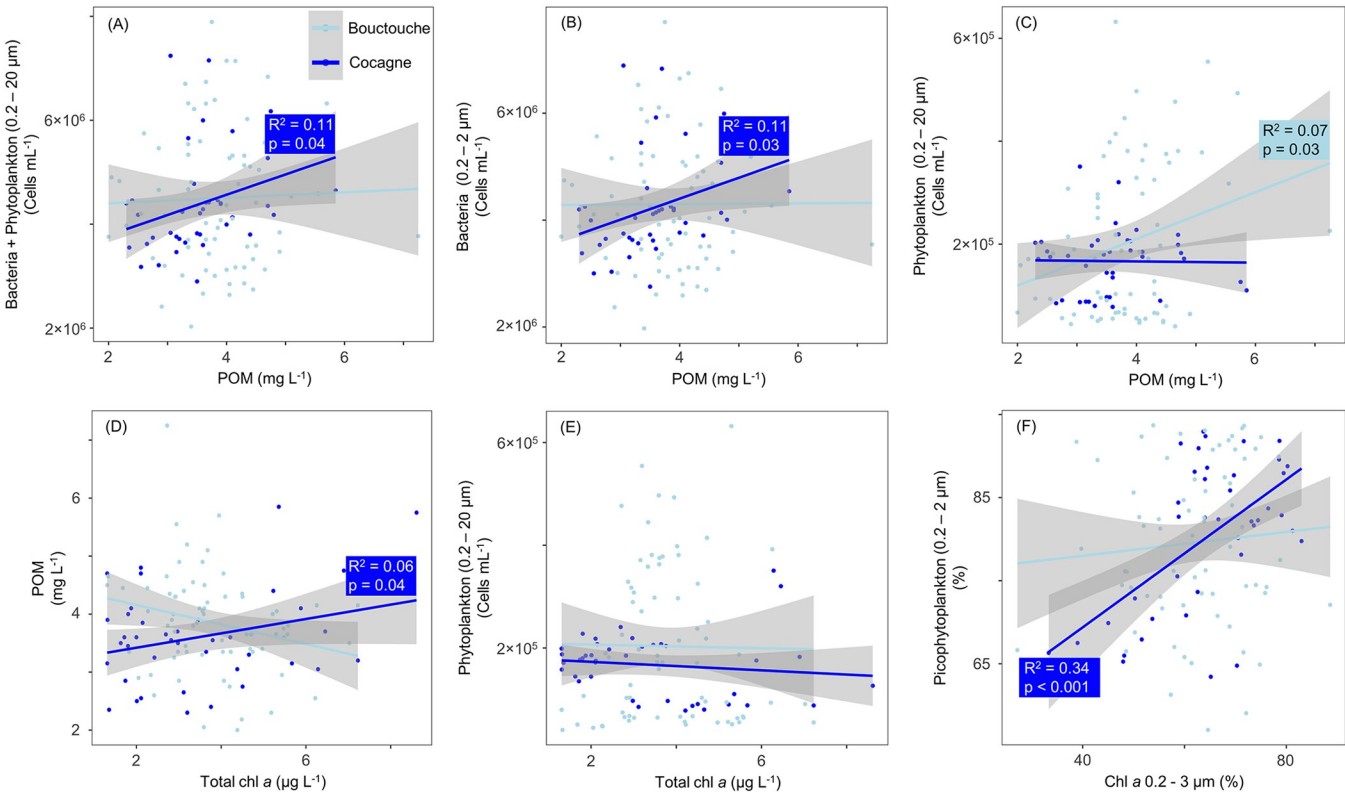

**Fig 11. Linear regressions comparing abundance and biomass data from Bouctouche and Cocagne.** (A) particulate organic matter (POM) vs total plankton (0.2 to 20 μm) abundance, (B) POM vs bacteria abundance, (C) POM vs phytoplankton abundance, (D) total chl *a* vs POM, (E) total chl *a* vs phytoplankton abundance, and (F) relative biomass of chl *a* 0.2 to 3 μm (of total chl *a*) vs picophytoplankton relative abundance. Only the $R^2$ and p values of statistically significant relationships are shown.

Agricultural land use in the New Brunswick watersheds (12 and 8% for Bouctouche and Cocagne, respectively; DFO unpublished data) also likely contributed more nutrients than in Nova Scotia embayments, < 1% in Country Harbour and Whitehead [68]. The shallow embayments may also be more susceptible to resuspension of seafloor sediments during high-wind events, which may re-introduce nutrients to the water column. Additionally, varying oceanic influence may create differences, as the deeper Nova Scotia embayments open to the Atlantic Ocean, whereas the shallow New Brunswick embayments open to the Northumberland Strait, which is a somewhat enclosed, shallow coastal area in the Gulf of St. Lawrence, influencing water temperatures which are warmer in the Northumberland Strait than the open Atlantic Ocean. A difference in water sources was also observed at Country Harbour (Fig 2B), where different phytoplankton composition was measured between riverine-influenced samples (Inner station at high and low tide as well as Mid station at low tide) and marine-influenced samples (Outer station at high and low tide as well as Mid station at high tide).

## Seasonal effects

Temperate bivalve aquaculture embayments are influenced by strong seasonality and variation in marine environmental conditions (e.g., water temperature, terrestrial freshwater and nutrient input, sea ice cover, water column stratification, light availability). Previous studies have shown that bay-specific characteristics such as local hydrodynamics drive environmental factors for bacteria and phytoplankton abundances. For example, seasonal plankton succession in

a deep (20 to 50 m) Canadian bivalve aquaculture embayment (South Arm) on the northeast coast of Newfoundland is driven by water column stratification [69]. Timing and intensity of stratification are generally influenced by freshwater input and atmospheric forcing (i.e. air temperature and wind) and ultimately affect nutrient and light availability, which are the most important variables controlling phytoplankton growth in subarctic marine systems [70, 71]. In the present study, seasonality was investigated in much shallower embayments (< 10 m) and well-mixed water columns, such that stratification may not be the driving force for plankton growth. Inorganic nutrient measurements were obtained from Bouctouche (S1 Fig) in conjunction with water collection for flow cytometry. Results suggest that nitrogen availability may be the limiting factor for plankton growth at Bouctouche, as low nitrate and nitrite levels were observed during the summer, likely due to depletion by plankton uptake, whereas phosphate and silicate remained available in the water column.

Heterotrophic bacteria (here including both bacteria and archaea) and phytoplankton abundance (Figs 4 and 6) in marine environments are highly interconnected and often are mutually beneficial [72]. This is because phytoplankton cell lysis and excretion are used as a source of organic matter for bacteria, which subsequently re-mineralize organic matter, providing nutrients for phytoplankton. However, both groups compete for the same pool of nutrients and phytoplankton typically dominate when fresh supplies of nutrients are present, whereas bacteria growth often peaks following maximal phytoplankton production and dominates in oligotrophic waters [73–75]. This pattern was observed in Bouctouche (sampled in 2022; Fig 8) whereas bacteria abundance peaked first, followed by phytoplankton abundance in Cocagne and South Arm (both embayments sampled in 2021). Overall, the timing in peak phytoplankton abundance was shifted later by a month at South Arm (September), compared to Bouctouche and Cocagne (August), likely due to differences in latitude, water column depth and stratification, and oceanic influence (i.e., Newfoundland shelf vs Northumberland Strait in southern Gulf of Saint- Lawrence).

In addition to intrinsic differences in hydrodynamics and river influence, differences in the timing, magnitude, and plankton composition between Cocagne and Bouctouche may have been caused by climatic differences between the two sampling years (Cocagne in 2021, Bouctouche in 2022). The year 2022 reached record high sea-surface temperatures (since time-series began in 1991) and warmed to summer temperatures two-weeks earlier than 2021 [76, 77]. Additionally, the month of August experienced 175% (Government of Canada Historical Climate Data, Kouchibouguac station) more precipitation in 2022 (181.9 mm) compared to 2021 (66.2 mm), which may have enhanced nutrient availability from terrestrial runoff in 2022, contributing to a much higher peak in phytoplankton abundance at Bouctouche, which was dominated by picoPC-cyanobacteria (Fig 6), a freshwater indicator [34]. The differences observed in bacteria and plankton communities between these two embayments may therefore be driven by interannual climatic differences rather than, or in addition to, aquaculture impacts (bivalve grazing pressure, nutrient alterations, benthic-pelagic coupling). This observation reinforces that inter-bay comparisons require a monitoring design that overcomes natural seasonal stochasticity. This may be accomplished by capturing seasonality when monitoring inter-year changes (e.g., sampling at multiple periods) or limiting inter-year variation when contrasting ecosystems between embayments (e.g., sampling the same year at the same periods).

## Organic resources and ecosystem functioning

Plankton abundance and relative abundance data are important for informing community structure and biodiversity, as well as having the ability to be correlated to changes in

environmental conditions [39, 40]. However, they are not directly representative of the food and energy availability to higher trophic levels, as biomass and nutritional content varies between taxonomic groups. Information on plankton biomass, which is often estimated as a function of carbon content, is thus essential to better understand and model aquaculture ecosystem interactions and implications for higher trophic levels. The average carbon content of the cells analyzed using flow cytometry spans several orders of magnitude, with coastal heterotrophic bacteria estimated at 0.03 pg C cell$^{-1}$ [78] and cyanobacteria and picoeukaryotes estimated at 0.1 and 1.5 pg C cell$^{-1}$, respectively [79, 80]. Estimating carbon content of nanoeukaryotes is more challenging as it depends on biovolume [81, 82], which must account for highly variable cell size and shape. However, with estimates exceeding 100 pg C cell$^{-1}$, nanoeukaryotes often dominate cell biomass < 20 μm, even when their relative abundance is low.

Biomass is therefore an important consideration to interpret plankton seasonality and timing of the summer phytoplankton bloom in embayments such as Bouctouche, which showed a strong seasonal pattern. Phytoplankton abundance was greatest in August due to high numbers of picoplankton, but phytoplankton biomass may have been greatest in July due to the increased contribution of nanoeukaryotes (> 73% picophytoplankton in July, > 86% picophytoplankton in August). Phytoplankton biomass may also be estimated using bulk filtrations of POM and chl *a*, using filters of different porosity to quantify size classes (pico, nano, micro). However, the lack of relationship between chl *a* and POM concentrations, as well as chl *a* and phytoplankton abundance (Fig 11D and 11E), indicate that organic pools at Bouctouche were not driven by phytoplankton < 20 μm, but likely had significant contributions from other marine and terrestrial plants (e.g., eelgrass beds), resuspended organic matter, fecal matter, or larger plankton. Microscopy counts or imaging technologies could additionally be used to quantify nano- and micro-phytoplankton and their biovolume for continued understanding of ecosystem functioning.

## Nutrient effects

Picophytoplankton composed 40 to 80% of total autotrophic biomass in enclosed shallow embayments (Bouctouche and Cocagne; Fig 9E). Using data compiled from 38 published reports spanning a wide range of environments, Agawin et al. [83] determined that picophytoplankton generally contribute > 50% of autotrophic biomass in warm (> 26˚C), nutrient-poor (nitrate + nitrite < 1 μM) waters. Therefore, the significant proportions of picophytoplankton observed here may be due to local nitrate and nitrite depletion (S1 Fig). Smaller phytoplankton cells perform better and tend to dominate under nutrient-limited conditions due to their higher surface area to volume ratio, which allows for more efficient nutrient acquisition [84, 85]. Previous studies conducted in aquaculture embayments have reported increased relative abundance [86] and biomass [2, 3, 87] of picophytoplankton due to increased ingestion efficiency of cells > 4 μm during bivalve filtration [12–15]; however, this trend was not observed here. In fact, within-bay comparisons of stations showed no significant difference in picophytoplankton biomass between stations at Bouctouche, while at Cocagne, the North station had significantly lower picophytoplankton biomass, mainly driven by low biomass recorded in June (Figs 9E and 10E). In conjunction, relative abundances of picophytoplankton, determined by flow cytometry, were lowest at the North and Inner stations within all embayments (Figs 5D and 8D). We therefore hypothesize that direct bivalve grazing, which would have increased picophytoplankton, is not the primary driver of picophytoplankton spatial patterns within the embayments studied here, but rather nutrient limitation, particularly nitrate and nitrite.

Notably, bivalve aquaculture sites may serve as either net sinks or sources of nitrogen depending on hydrodynamics, farming practices, sediment conditions, sediment resuspension rates and strength of benthic-pelagic coupling [88–90]. Benthic-pelagic coupling may reintroduce remineralized nutrients from sediments into the pelagic ecosystem and stimulate primary production [91]. Moving forward, results highlight the importance of assessing spatio-temporal variation of inorganic nutrients in conjunction with plankton to investigate ecosystem effects due to nutrient alterations within bivalve aquaculture embayments.

The contribution of HNA bacterial cells to total heterotrophic bacteria was additionally measured as a proxy for nutrient conditions, which showed the highest HNA proportions, observed in the shallow enclosed embayments, near-lease in late spring (May, June) and late summer (August, September), as well as near riverine inputs. These results showed high spatio-temporal variability, particularly as a function of seasonality, limiting the assessment of aquaculture effects and emphasizing the need for higher spatial resolution (e.g. under and near-lease) and replication. Whereas HNA determination from flow cytometry may be better used as a rapid and cost-efficient indicator method to observe potential long-term bay-scale changes in bacterial communities, molecular methods such as metabarcoding and quantitative PCR would allow a more in-depth characterization of bacteria composition and determination of functional genes and thus better inform biogeochemical cycling processes occurring near-lease [92–94].

## Aquaculture interactions

Country Harbour is the only embayment in the study with no bivalve aquaculture grazing pressure due to empty leases at the time of sampling, and total phytoplankton cell abundance decreased from the Inner to Outer stations (Fig 5). This aligns with previous studies done in coastal bays, which have reported an inner to outer bay decrease in summer phytoplankton abundance along freshwater and nutrient gradients [95–97]. In contrast to Country Harbour, the three other embayments, which contained several active leases, showed no significant difference in phytoplankton abundance along the bay gradient. We hypothesize that the relatively greater number of oyster leases at the Inner (Whitehead) and North (Bouctouche, Cocagne) stations may lead to increased consumption of phytoplankton cells, thereby reducing phytoplankton abundance and altering the natural inner to outer bay gradient. Additionally, hydrodynamic models have shown that the North stations at the shallow enclosed embayments have the highest water residence times (Bouctouche: 60.3 and 49.7 days at North and Central stations, respectively. Cocagne: 20.2, 14.4 and 19.2 days at North, Central and South stations, respectively; Guyondet, unpublished). This may decrease the rate at which additional phytoplankton cells are replenished and increase the vulnerability of these areas to aquaculture grazing pressure. Long-term monitoring through changes in aquaculture production within selected embayments may allow successful differentiation between spatially confounding factors from aquaculture (grazing, nutrient alteration and benthic-pelagic coupling) and natural coastal dynamics (water residence time, nutrient inputs and plankton uptake).

Within-bay patterns additionally showed that the stations most influenced by leases had the highest bacteria abundance, supporting that bacteria growth may be promoted by aquaculture activity due to increased organic nutrient availability through both dissolved and particulate excretion [98, 99]. Further studies are required to investigate if bacterial remineralization may enhance local inorganic nutrient availability for phytoplankton growth and auto-regulate primary production in relation to aquaculture level and local dynamics, and/or if long-term nutrient alteration and energy redirection towards the microbial loop may ultimately limit energy transfer to higher pelagic trophic levels [100–102].

### Recommendations for monitoring

Here, we provide an overview of factors to be considered when monitoring bacteria and phytoplankton at the bay-scale in aquaculture embayments. Although the proportion of picophytoplankton has previously been proposed as an indicator of bivalve grazing, we observed that spatio-temporal trends in picophytoplankton were driven by nutrient gradients, limiting the ability to disentangle bivalve grazing pressure and aquaculture-related nutrient alterations from other spatial and seasonal variation. To avoid misinterpreting ecosystem changes as a result of aquaculture, seasonality must be considered with sampling seasons spanning at least 3 to 4 months [69] to capture plankton dynamics and account for seasonal phenology shifts expected from climate change, and variability of grazing rates. To assess bay-scale composition, other local factors such as tide phase should be accounted for in areas where riverine vs marine influence may affect phytoplankton composition. Additionally, where the water column is stratified, water samples should be collected at multiple depths for a comprehensive understanding of plankton dynamics. To successfully detect long-term bay-scale changes in relation to aquaculture and differentiate confounding factors, we recommend using a multi-metric approach to assess and forecast holistic bay-scale changes of ecosystem functioning. In addition to flow cytometry abundance data, plankton biomass is important for informing food and energy resources for higher trophic levels, and may be estimated using bulk filtrations of POM and chl *a*, using filters of different porosity to quantify size classes (pico, nano, micro). Additionally, species determination of cells > 20 μm should be conducted using FlowCam imaging or traditional microscopy to quantify the full spectrum of aquaculture grazing. Finally, disentangling the effect of aquaculture pressure from stochastic processes, eutrophication, climate change, and fisheries would require appropriate reference sites. Assessing and forecasting aquaculture management thresholds require a better understanding of aquaculture production levels and will further require monitoring ecosystem changes over a gradient of aquaculture production in similar ecosystems (i.e., plankton composition, hydrodynamic regime) to reduce effects from confounding factors [103].

### Conclusions

There is a need to better understand aquaculture ecosystem interactions and to develop predictive carrying capacity models to avoid a state where the role and functions of plankton would be substantially altered by cultured bivalve dominance. Here, empirical results show that bivalve grazing may reduce near-lease phytoplankton abundance and favor bacterial growth. However, it is unknown if ecosystem functioning is altered at the bay-scale. For example, it remains unclear if near-lease benthic-pelagic coupling may enhance bacterial remineralization of organic nutrients, made available through dissolved and particulate excretion from cultured bivalves, and auto-regulate primary production and/or limit energy transfer to higher trophic levels. Overall, environmental forcings appeared to be the main drivers of spatio-temporal trends in phytoplankton composition, which showed strong variation as a function of riverine vs marine influence and nutrient limitations. Therefore, a long-term aquaculture monitoring program assessing planktonic communities must be carefully designed to be able to disentangle natural and anthropogenic factors to adequately support management decision-making.

### Supporting information

**S1 Fig. Inorganic nutrients concentrations from Bouctouche.** (A) Phosphate, (B) nitrite and nitrate, (C) nitrate and (D) silicate. Water samples for inorganic nutrient concentrations were collected at the same time and locations as water samples for flow cytometry, however stations

are unknown here.
(TIF)

## Acknowledgments

Thanks to the Aquaculture Monitoring Program (AMP) working group members, the Advisory Committee, and program management for exchanging scientific ideas on this project, including Johannie Duhaime who coordinated AMP. Thank you to Lara Cooper for help with funding acquisition and program management. Thank you to Ramón Filgueira for his help with sampling design and logistics. Khang Hua for reviewing sampling protocol and compiling and providing the data and metadata. Thank you to Stephen Finnis for metadata compilation and guidance with statistical analysis and R scripts. We thank Brian Fortune for his support to access the Country Harbour and Whitehead sites, logistics, and aquaculture information. Thank you to Julie Arseneau and Steven Neil for their help with sample collection at Country Harbour and Whitehead, and to Jeffery Clements, Johannie Duhaime, Michael Coffin, Katherine Gingles, Sarah Harrison, Julie LeRay, Rémi Sonier, and Véronic St-Laurent for assistance with fieldwork at Bouctouche and Cocagne.

## Author Contributions

**Conceptualization:** Thomas Guyondet, Jeffrey Barrell, Christopher W. McKindsey, Flora Salvo, Anaïs Lacoursière-Roussel.

**Data curation:** Jeffrey Barrell, Claude Belzile, Anaïs Lacoursière-Roussel.

**Formal analysis:** Hannah Sharpe, Anaïs Lacoursière-Roussel.

**Funding acquisition:** Thomas Guyondet, Jeffrey Barrell, Christopher W. McKindsey, Anaïs Lacoursière-Roussel.

**Investigation:** Jeffrey Barrell, Anaïs Lacoursière-Roussel.

**Methodology:** Thomas Guyondet, Claude Belzile, Flora Salvo, Anaïs Lacoursière-Roussel.

**Project administration:** Thomas Guyondet, Jeffrey Barrell, Anaïs Lacoursière-Roussel.

**Supervision:** Anaïs Lacoursière-Roussel.

**Visualization:** Hannah Sharpe.

**Writing – original draft:** Hannah Sharpe.

**Writing – review & editing:** Hannah Sharpe, Thomas Guyondet, Jeffrey Barrell, Claude Belzile, Christopher W. McKindsey, Flora Salvo, Anaïs Lacoursière-Roussel.

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
