## [Decision Letter · Decision Letter 0]

1 Sep 2024

PONE-D-24-31966Monitoring bay-scale bivalve aquaculture ecosystem interactions using flow cytometryPLOS ONE

Dear Dr. Sharpe,

Thank you for submitting your manuscript to PLOS ONE. After careful consideration, we feel that it has merit but does not fully meet PLOS ONE’s publication criteria as it currently stands. Therefore, we invite you to submit a revised version of the manuscript that addresses the points raised during the review process.

We look forward to receiving your revised manuscript.

Kind regards,

Qing Wang

Academic Editor

PLOS ONE

Journal Requirements:

2. Thank you for stating the following financial disclosure: This project was funded by the Fisheries and Oceans Canada Aquaculture Monitoring Program. 

Reviewers' comments:

Reviewer's Responses to Questions

**Comments to the Author**

1. Is the manuscript technically sound, and do the data support the conclusions?

Reviewer #1: Partly

Reviewer #2: Yes

Reviewer #3: Partly

2. Has the statistical analysis been performed appropriately and rigorously? 

Reviewer #1: Yes

Reviewer #2: Yes

Reviewer #3: Yes

3. Have the authors made all data underlying the findings in their manuscript fully available?

Reviewer #1: Yes

Reviewer #2: Yes

Reviewer #3: Yes

4. Is the manuscript presented in an intelligible fashion and written in standard English?

Reviewer #1: Yes

Reviewer #2: Yes

Reviewer #3: Yes

5. Review Comments to the Author

Reviewer #1: Review of PONE-D-24-31966

This study used flow cytometry to investigate natural spatial and temporal drivers of plankton dynamics in coastal embayment’s with shellfish aquaculture leases. Although environmental forcing appeared to be a strong driver of plankton dynamics there was some evidence that bivalve aquaculture may reduce near-lease phytoplankton abundance and favor bacterial growth.

General comments

I want to first acknowledge that this paper is not exactly within my areas of expertise so please take my comments accordingly.

I accepted the review because I was interested in the topic with regards to the impacts of shellfish aquaculture on marine nearshore habitats. Although the first two paragraphs of the introduction focused on shellfish aquaculture including effects on plankton dynamics that may affect other commercially important species like lobster, the effect of shellfish aquaculture on plankton dynamics was only tangentially assessed.

Given this, I suggest the authors re-work their paper to emphasize their study addresses some of the environmental drivers that may confound investigating the effect of shellfish aquaculture on plankton dynamics, as they didn’t explicitly test for a shellfish effect.

Reviewer #2: This manuscript investigates the spatio-temporal variations in bacteria (i.e., bacteria and archaeal communities) and pico- and nanophytoplankton composition in four contrasted bivalve aquaculture embayments using flow cytometry.The proposed work is a solid attempt to show that bivalve aquaculture may reduce near-lease phytoplankton abundance and favor bacterial growth, even if environmental forcings appeared to be a strong driver of spatio-temporal trends of these communities. In addition, this manuscript provides an interesting and complete discussion on aquaculture monitoring and their effects on more globally benthic-pelagic coupling processes and microbial biogeochemical cycling.

Overall, this manuscript is well written and is an interesting contribution that is wholly suitable with minor revisions for publication in PlosOne. There are indeed few areas of the manuscript that could be improved or clarified.

First, the number of items (13 figures + 2 tables) in the main text is a bit disproportional. Not all the figures are needed to be placed in the main text and the authors may consider to move some of them as supplementary material or detail them a bit more than in the current version of the manuscript in the results section, or even redo some of them. For instance, the three NMDS plots could be eventually placed in a single figure.

L87-90: Another non-negligeable limit of the technics is the low taxonomic resolution that can be obtained from it. With flow cytometry, researchers can only quantify bacteria and phytoplankton at a wide taxonomic level (i.e., assemblages based on size, pigmentation or level of nucleic acid content) while it has been shown for instance that bivalves may alter the relative biomass of certain phyla/species such as diatoms by their selective grazing activity (Lucas et al., 2016, doi: 10.3389/fmars.2016.00014).

L190: Was there a prefiltration on 20 um to only catch and measure the < 20 um planktonic community?

L220-222: The authors have studied the planktonic community composition (abundance of planktonic taxa) using flow cytometry. It is yet unclear what the authors meant by taxa here (which suppose a taxonomic resolution close to the genus level) and how they have achieved this taxonomic identification thanks to the glow cytometry scatters they obtained... More details need to be given before in the M&M section on how the authors have categorized the studied planktonic community at the taxa level. However, if by “taxa” the authors meant, for instance for bacteria, the differentiation between HNA and LNA, the term “assemblage” will be more adapted throughout the text, I guess.

L255-258: Please specify in the legend caption the bacterial and phytoplanktonic composition were assessed by flow cytometry. As mentioned above it is unclear what the authors meant by “community composition” and what are the “taxa” they have considered in their analysis. It is then difficult to judge on the pertinence of the NMDS analysis done in that figure.

Reviewer #3: Review of PONE-D-24-31966

This article reports on spatio-temporal variability in planktonic bacteria and phytoplankton communities in 4 Eastern Canadian embayments with a link with aquaculture intensity. The ultimate goal of the article is to inform on proper sampling design to detect impacts of aquaculture activities on bay-scale pelagic primary producers. I feel the article is well written (I added several minor comments and edits in the manuscript file attached), is easy to follow and the statistical analyses and result presentation are generally adequate. However, I have more severe criticism about the sampling design and the link between the ultimate objective and how the study was designed.

First, I appreciate the challenges of designing a sampling scheme that allows comparisons among very different environments. It is clear that one cannot sample plankton at 3 different depths if the water depth does not allow it. However, the variations in the sampling design among the different embayments reported here is huge (different number of samples, different number of sampling dates, different years, different sampling locations in relation to position of leases, different depths, different tidal cycles, and different variables measured). In my opinion, it would be important to discuss why this final design was chosen and why, for example, it was decided not to take routine measurements (such as Chl a, POM, SPM in Nova Scotia) or sample the two New Brunswick sites on two different years rather than simultaneously.

From the last two sentences of the abstract, I was expecting the discussion to focus on details pertaining to aquaculture impacts and how to detect them. I found that a large part of it instead attempted to identify causes (other than aquaculture) for the patterns observed. In my opinion, the discussion should be framed to discuss, in light of the levels of spatio-temporal variability observed in the present study, how do we move forward to better understand the impact of aquaculture on ecosystem functioning using monitoring programs. There are parts of the discussion that do so, but in my opinion the topic could be expanded to better fit the stated aim of the article. I see value in the data presented in terms of a baseline to build on to come up with a proper design to detect bay-scale effects if they indeed exist. Reading the title and the abstract, I was expecting the article to describe a sampling design, that uses flow cytometry, that was capable of separating the farm effects from all other factors shaping planktonic communities. Clearly, this is not the case, and the article could be re-framed with a ‘lessons learned’ format.

I have on concern about the statistical analyses. From my understanding, Mann-Withney U tests are pairwise tests. When comparing multiple levels of a factor (for example, multiple dates within an embayment) the type-1 error rate is inflated. I think a Bonferonni correction or others methods should be used to keep the error rate constant.

6. PLOS authors have the option to publish the peer review history of their article (what does this mean?). If published, this will include your full peer review and any attached files.

Reviewer #1: No

Reviewer #2: No

Reviewer #3: No

---

## [Author Response · Author response to Decision Letter 0]

3 Oct 2024

Dear Reviewers, 

We thank you for your time and thoughtful comments. After incorporating your suggestions, we feel the manuscript has been significantly improved. Below, please find our detailed answers to your comments. Please note that all line and figure numbers referred to below have been changed to align with the new version of the manuscript.

Sincerely, 

Hannah Sharpe, on behalf of all co-authors

Reviewer #1: 

This study used flow cytometry to investigate natural spatial and temporal drivers of plankton dynamics in coastal embayment’s with shellfish aquaculture leases. Although environmental forcing appeared to be a strong driver of plankton dynamics there was some evidence that bivalve aquaculture may reduce near-lease phytoplankton abundance and favor bacterial growth. 

I want to first acknowledge that this paper is not exactly within my areas of expertise so please take my comments accordingly.

I accepted the review because I was interested in the topic with regards to the impacts of shellfish aquaculture on marine nearshore habitats. Although the first two paragraphs of the introduction focused on shellfish aquaculture including effects on plankton dynamics that may affect other commercially important species like lobster, the effect of shellfish aquaculture on plankton dynamics was only tangentially assessed.

Given this, I suggest the authors re-work their paper to emphasize their study addresses some of the environmental drivers that may confound investigating the effect of shellfish aquaculture on plankton dynamics, as they didn’t explicitly test for a shellfish effect.

Sharpe et al.:

We thank the reviewer for their comment and for contributing to improving the manuscript. We realize now that the title of the manuscript (Monitoring bay-scale bivalve aquaculture ecosystem interactions using flow cytometry) may confuse readers as to what we discuss throughout the paper. We have therefore changed the title of the manuscript to “Monitoring bay-scale ecosystem changes in bivalve aquaculture embayments using flow cytometry” to add clarity to the direction of the paper. 

We agree with Reviewer 1 that aquaculture effects should be better studied in the literature and our work is providing information required to move this field away from tangentially assessing aquaculture effects, and towards direct understanding of ecosystem changes as a result of aquaculture. We have attached a new version of the manuscript, with incorporated comments from Reviewers 2 and 3, and now better clarify the importance of understanding environmental forcings when monitoring and interpreting aquaculture sites. 

Reviewer #2: 

This manuscript investigates the spatio-temporal variations in bacteria (i.e., bacteria and archaeal communities) and pico- and nanophytoplankton composition in four contrasted bivalve aquaculture embayments using flow cytometry.The proposed work is a solid attempt to show that bivalve aquaculture may reduce near-lease phytoplankton abundance and favor bacterial growth, even if environmental forcings appeared to be a strong driver of spatio-temporal trends of these communities. In addition, this manuscript provides an interesting and complete discussion on aquaculture monitoring and their effects on more globally benthic-pelagic coupling processes and microbial biogeochemical cycling.

Overall, this manuscript is well written and is an interesting contribution that is wholly suitable with minor revisions for publication in PlosOne. There are indeed few areas of the manuscript that could be improved or clarified.

Sharpe et al.:

We thank the reviewer for their positive comments and for contributing to improving the manuscript. Please note that the line numbers refer to the revised version of the manuscript. In addition to the revised manuscript, we have uploaded a tracked changes version in order to make the changes stand out.

Reviewer #2:

First, the number of items (13 figures + 2 tables) in the main text is a bit disproportional. Not all the figures are needed to be placed in the main text and the authors may consider to move some of them as supplementary material or detail them a bit more than in the current version of the manuscript in the results section, or even redo some of them. For instance, the three NMDS plots could be eventually placed in a single figure.

Sharpe et al.:

Following the reviewer’s suggestion, we have combined the three NMDS figures into one (Figure 2). We feel that the remaining figures are important to easily observe and interpret the inter- and within-bay patterns discussed in the text. We thus believe there is value to keeping the remainder of the figures in the main text. 

Reviewer #2:

L87-90: Another non-negligeable limit of the technics is the low taxonomic resolution that can be obtained from it. With flow cytometry, researchers can only quantify bacteria and phytoplankton at a wide taxonomic level (i.e., assemblages based on size, pigmentation or level of nucleic acid content) while it has been shown for instance that bivalves may alter the relative biomass of certain phyla/species such as diatoms by their selective grazing activity (Lucas et al., 2016, doi: 10.3389/fmars.2016.00014).

Sharpe et al.:

We thank the reviewer for this comment and agree. Following the reviewer’s suggestion, we have thus revised lines 88-98 to read:

“While abundance and relative abundance informs community structure and biodiversity, and shows better correlation to variations in environmental conditions (39,40), a limit of flow cytometry is the ability to accurately assess the biovolume, and therefore biomass, of the cells analyzed. Additionally, the low taxonomic resolution obtained from flow cytometry only allows for bacteria and phytoplankton quantification based on particle size, pigmentation, or level of nucleic acid content. This is a limitation of the method since bivalves may alter the relative biomass of certain phyla/species by selective grazing (41). Biomass and species-specific data therefore have important implications for ecosystem productivity and food availability to higher trophic levels, and must be measured using additional methods such as bulk filtrations, microscopy counts, or imaging technology.”

Reviewer #2:

L190: Was there a prefiltration on 20 um to only catch and measure the < 20 um planktonic community?

Sharpe et al.:

No, the samples were not prefiltered. Most flow cytometers, including the one used in this study, do not detect cells larger than 20 µm because of the way the water sample is focused in a 20 µm stream inside the flowcell where the particles interact with the laser photons. 

Reviewer #2:

L220-222: The authors have studied the planktonic community composition (abundance of planktonic taxa) using flow cytometry. It is yet unclear what the authors meant by taxa here (which suppose a taxonomic resolution close to the genus level) and how they have achieved this taxonomic identification thanks to the glow cytometry scatters they obtained... More details need to be given before in the M&M section on how the authors have categorized the studied planktonic community at the taxa level. However, if by “taxa” the authors meant, for instance for bacteria, the differentiation between HNA and LNA, the term “assemblage” will be more adapted throughout the text, I guess.

Sharpe et al.:

We thank the reviewer for their comment. We agree that the term “assemblage” is better-fitting than “taxa”. In an attempt to be as clear as possible, we have edited lines 231-234 to read: 

“Two-dimensional non-metric multidimensional scaling (NMDS) ordinations (61) were used to graphically display spatio-temporal patterns in bacteria (HNA, LNA) and phytoplankton (picoPC-cyanobacteria, nanoPC-cyanobacteria, picoPE-cyanobacteria, nanoPE-cyanobacteria, picoeukaryotes, nanoeukaryotes) community composition (assemblage and abundance).”

Reviewer #2:

L255-258: Please specify in the legend caption the bacterial and phytoplanktonic composition were assessed by flow cytometry. As mentioned above it is unclear what the authors meant by “community composition” and what are the “taxa” they have considered in their analysis. It is then difficult to judge on the pertinence of the NMDS analysis done in that figure.

Sharpe et al.:

We have added in the figure caption that these data are assessed using flow cytometry. We now feel it is clearer what we mean by taxa (now referred to as assemblages), as addressed in the previous comment. 

Reviewer #3: 

This article reports on spatio-temporal variability in planktonic bacteria and phytoplankton communities in 4 Eastern Canadian embayments with a link with aquaculture intensity. The ultimate goal of the article is to inform on proper sampling design to detect impacts of aquaculture activities on bay-scale pelagic primary producers. I feel the article is well written (I added several minor comments and edits in the manuscript file attached), is easy to follow and the statistical analyses and result presentation are generally adequate. However, I have more severe criticism about the sampling design and the link between the ultimate objective and how the study was designed.

Sharpe et al.:

We thank the reviewer for their positive comments and for contributing to improving the manuscript. Please note that the line numbers refer to the revised version of the manuscript. In addition to the revised manuscript, we have uploaded a tracked changes version in order to make the changes stand out.

Reviewer #3:

First, I appreciate the challenges of designing a sampling scheme that allows comparisons among very different environments. It is clear that one cannot sample plankton at 3 different depths if the water depth does not allow it. However, the variations in the sampling design among the different embayments reported here is huge (different number of samples, different number of sampling dates, different years, different sampling locations in relation to position of leases, different depths, different tidal cycles, and different variables measured). In my opinion, it would be important to discuss why this final design was chosen and why, for example, it was decided not to take routine measurements (such as Chl a, POM, SPM in Nova Scotia) or sample the two New Brunswick sites on two different years rather than simultaneously.

Sharpe et al.:

Operational and logistical constraints are one of the explanations for the resulting datasets in each system. However, clear patterns could be found at the bay-scale (e.g. Figure 2). Although the inconsistencies limited the comparison between systems, this dataset is rather used to test factor effect on bacteria and phytoplankton composition by 1) taking advantage of the system differences to test specific factors where they can be tested, and 2) for factors where sampling consistency is sufficient, we evaluate if similarities emerged across different systems.

Reviewer #3:

From the last two sentences of the abstract, I was expecting the discussion to focus on details pertaining to aquaculture impacts and how to detect them. I found that a large part of it instead attempted to identify causes (other than aquaculture) for the patterns observed. In my opinion, the discussion should be framed to discuss, in light of the levels of spatio-temporal variability observed in the present study, how do we move forward to better understand the impact of aquaculture on ecosystem functioning using monitoring programs. There are parts of the discussion that do so, but in my opinion the topic could be expanded to better fit the stated aim of the article. I see value in the data presented in terms of a baseline to build on to come up with a proper design to detect bay-scale effects if they indeed exist. Reading the title and the abstract, I was expecting the article to describe a sampling design, that uses flow cytometry, that was capable of separating the farm effects from all other factors shaping planktonic communities. Clearly, this is not the case, and the article could be re-framed with a ‘lessons learned’ format.

Sharpe et al.:

We thank the reviewer for their comment and realize now that the manuscript title and abstract may confuse readers as to what we discuss throughout the paper. We have therefore changed the title of the manuscript to “Monitoring bay-scale ecosystem changes in bivalve aquaculture embayments using flow cytometry” to add clarity to the direction of the paper. We additionally edited the last two sentences of the abstract to increase clarity: “We discuss the confounding environmental factors that must be accounted for when interpreting aquaculture effects such as grazing, benthic-pelagic coupling processes, and microbial biogeochemical cycling. Conclusions provide guidance on sampling considerations using flow cytometry in aquaculture sites based on embayment geomorphology and hydrodynamics.” (lines 34-38).

Reviewer #3:

I have on concern about the statistical analyses. From my understanding, Mann-Withney U tests are pairwise tests. When comparing multiple levels of a factor (for example, multiple dates within an embayment) the type-1 error rate is inflated. I think a Bonferonni correction or others methods should be used to keep the error rate constant.

Sharpe et al.:

A paired t-test determines whether the mean change for pairs is significantly different from zero. Comparatively, the Mann Whitney U test is a nonparametric hypothesis test that compares two independent groups using a ranking procedure and quantifies if the two groups are statistically similar or different (i.e., the difference between the medians is different and/or the groups have different distributions). This test is best used when you want to compare two independent groups, the data follow a non-normal distribution, and sample sizes are small (<30). The Mann Whitney U test is therefore not a pairwise test and is appropriate for our dataset. 

One important benefit to using the Mann-Whitney U test is that it can be used when comparing unequal sample sizes, which is the case for several of our applications (Fig 3: late summer inter-bay comparisons; Figs 8 and 10: monthly comparisons in Bouctouche, and station comparisons in Cocagne). 

● https://typeset.io/questions/is-mann-whitney-u-test-applicable-if-sample-sizes-for-ji0aodos8j

● https://doi.org/10.1002/bimj.201600022

As the reviewer mentions, it is possible that the type I error is amplified in a situation of heteroscedasticity (unequal scatter of residuals). 

● https://www.tqmp.org/RegularArticles/vol04-1/p013/p013.pdf

● Mann-Whitney U test when variances are unequal (pennds.org)

We therefore conducted the Breusch Pagan Test for comparisons with equal sample sizes, to test if our data is heteroscedastic, and we found that our data did not meet heteroscedasticity 97.7% of the time. This further confirms that the Mann-Whitney U test is appropriate for our dataset and the Bonferonni Correction is not necessary.

● https://www.statology.org/breusch-pagan-test-r/

● https://sscc.wisc.edu/sscc/pubs/RegDiag-R/homoscedasticity.html

We additionally conducted a Kruskal-Wallis Kruskal-Wallis test and performed the Dunn's post hoc pairwise comparison with the Bonferroni correction, to see if this would alter our manuscript. We found only slight differences between the two statistical methods, which did not change the discussion or conclusion of the manuscript. Please see below the same figures as shown in the manuscript, but using Kruskal-Wallis/Dunn/Bonferroni statistics rather than the Mann Whitney U Test (see figures below).

We have therefore decided that it would be best to keep the Mann Whitney U Test and, to increase clarity, we have added the following text to the manuscript: “Boxplots were used to visualize spatio-temporal patterns in bacteria and phytoplankton abundance, % HNA (to total bacteria), % picophytoplankton and % cyanobacteria (to total phytoplankton <20 μm). The first, second, and third quartiles, and lines extending from the boxes indicate the minimum and maximum values up to 1.5 times the interquartile range. To determine if each group differed statistically between each other for both medians and distribution, the non-parametric Mann-Whitney U test (wilcox.test

---

## [Decision Letter · Decision Letter 1]

22 Oct 2024

Monitoring bay-scale ecosystem changes in bivalve aquaculture embayments using flow cytometry

PONE-D-24-31966R1

Dear Dr. Sharpe,

We’re pleased to inform you that your manuscript has been judged scientifically suitable for publication and will be formally accepted for publication once it meets all outstanding technical requirements.

Kind regards,

Qing Wang

Academic Editor

PLOS ONE

Additional Editor Comments (optional):

Reviewers' comments:

Reviewer's Responses to Questions

**Comments to the Author**

1. If the authors have adequately addressed your comments raised in a previous round of review and you feel that this manuscript is now acceptable for publication, you may indicate that here to bypass the “Comments to the Author” section, enter your conflict of interest statement in the “Confidential to Editor” section, and submit your "Accept" recommendation.

Reviewer #2: All comments have been addressed

Reviewer #3: All comments have been addressed

2. Is the manuscript technically sound, and do the data support the conclusions?

Reviewer #2: Yes

Reviewer #3: Yes

3. Has the statistical analysis been performed appropriately and rigorously? 

Reviewer #2: Yes

Reviewer #3: Yes

4. Have the authors made all data underlying the findings in their manuscript fully available?

Reviewer #2: Yes

Reviewer #3: Yes

5. Is the manuscript presented in an intelligible fashion and written in standard English?

Reviewer #2: Yes

Reviewer #3: Yes

6. Review Comments to the Author

Reviewer #2: The authors have done a thorough work with the revised version and addressed and/or defended many of the issues that I have raised. I was pleased to see that they have followed most of my recommendation and I believe their manuscript has greatly gained in clarity. I am therefore pleased to recommend this revised version to be accepted for publication in PlosOne. Looking forward to see it online!

Reviewer #3: I feel that all comments from all reviewers have been adequately addressed and the manuscript is therefore suitable for publication

7. PLOS authors have the option to publish the peer review history of their article (what does this mean?). If published, this will include your full peer review and any attached files.

Reviewer #2: No

Reviewer #3: No

---

## [Editor Report · Acceptance letter]

26 Oct 2024

PONE-D-24-31966R1 

PLOS ONE

Dear Dr. Sharpe, 

I'm pleased to inform you that your manuscript has been deemed suitable for publication in PLOS ONE. Congratulations! Your manuscript is now being handed over to our production team.

Kind regards, 

on behalf of

Dr. Qing Wang 

Academic Editor

PLOS ONE